# τ-BEN: A Temporal Logic Guided Approach for Temporal Reasoning Benchmark Generation

## Abstract

In logic-based Artificial Intelligence, temporal reasoning typically involves formalizing problems as logical rule expressions and employing symbolic reasoners to infer and derive new conclusions from structured knowledge. However, symbolic reasoners cannot process natural language directly and require manually constructed symbolic knowledge bases, which can be both time-consuming and resource-intensive to create and maintain. Given the recent widespread adoption of Large Language Models (LLMs) and their remarkable successes across diverse domains, we are motivated to explore to what extent LLMs can handle temporal logic tasks, dispensing with traditional symbolic reasoners.

We introduce τ-BEN, a benchmark suite that adheres to the semantics of temporal logic. It synthesizes temporal reasoning datasets in both symbolic and natural language forms, enabling the evaluation of LLMs on temporal logic reasoning. τ-BEN is a highly scalable benchmark that supports the generation of datasets with varying sizes and rule structures of varying complexity. Furthermore, each question in τ-BEN is guaranteed to be unseen by LLMs during pretraining, effectively minimizing the risk of data leakage. Our results, along with a detailed ablation study of seven frontier LLMs, offer valuable insights into the capabilities and limitations of current models in temporal logic reasoning tasks.

## 1 Introduction

Temporal logic reasoning problems, grounded on formal logical rules, have been studied for decades in the field of logic-based Artificial Intelligence Alur & Henzinger (1994); Venema (2017); Lamport (1980). Predominant approaches to solving these problems typically rely on reasoners that are specific to particular logical languages, such as MeTeoR Wang et al. (2022) and NuSMV Cimatti et al. (1999). A drawback of employing symbolic reasoners for temporal logic reasoning is that they often require specialized knowledge bases and rules tailored to a specific temporal logic language, which can be both time-consuming and resource-intensive to create and maintain. Additionally, the inability to support natural language expressions also limits their applicability to other domains.

Currently, a widely adopted approach to calibrating the diverse capabilities of LLMs is the construction of well-designed and representative benchmarks. For example, HumanEval Chen et al. (2021) was introduced to evaluate the coding abilities of LLMs, while GSM8K Cobbe et al. (2021) was developed to assess their performance in mathematical reasoning. However, in traditional logic-based Artificial Intelligence (AI) domains, many tasks are still addressed using formal logical rules and symbolic reasoners. Despite the advancements of LLMs, relatively little effort has been made to explore their capabilities in solving such tasks—particularly the more challenging aspects of rule-based temporal logic reasoning. While some studies have benchmarked or evaluated the temporal reasoning abilities of LLMs Wang & Zhao (2024); Xiong et al. (2024), they primarily focus on reasoning over temporal data expressed in natural language, without addressing the temporal logic, which is typically represented as logical rules with well-established syntax and semantics.

In this paper, we introduce τ-BEN, a benchmark suite to evaluate the temporal reasoning capabilities of language models. Each question is constructed based on temporal logic and is guaranteed to be unseen during training, thereby requiring models to perform reasoning rather than rely on memorized knowledge. Specifically, we adopt DatalogMTL Brandt et al. (2018), a popular temporal logic language, as a proxy, and focus on the classic temporal logic reasoning task of *fact entailment* Cheng

(1996); Brandt et al. (2018). We consider temporal data of the symbolic form $P(a_1, \ldots, a_n)@\varrho$, where $P$ denotes a predicate (relation), $a_i$ is an entity, $n$ denotes the arity[1] and $\varrho$ represents a punctual time point or time interval. Given a set of temporal rules and a target temporal fact, the task is to determine whether the fact is entailed by the temporal data and logical rules. To provide better intuition, we use Example 1 togeter with Figure 1 to describe the problem.

**Example 1.** *There is growing evidence that individuals develop COVID-19 immunity if they were infected within the last 6 months (discounting the last ten days when they had no symptom) Feikin et al. (2022). The condition can be captured by a DatalogMTL program $\Pi_{ex}$ with the following rule:*

$$Immune(x) \leftarrow \diamondsuit_{(10,183]} Infect(x), \boxminus_{[0,10]} NoSym(x)$$

The above rule checks whether an individual infected at some point in the last six months excluding the last 10 days (operator $\diamondsuit_{(10,183]}$) remained continuously without symptoms in the last 10 days (using the 'box past' operator $\boxminus_{[0,10]}$).

Then, we assume a dataset contains some historical data about a person called Ben in the form of facts stamped with validity intervals, where the first day of the year is given by the interval $(0, 1]$, the second day by $(1, 2]$, and so on. Ben got vaccinated at July 19 (represented as 199). Moreover, Ben had no symptoms since July 1 (i.e., 181) until August 30 (i.e., 242). This is represented by a dataset $\mathcal{D}_{ex}$ with the following facts:

$$Infect(\text{Ben})@199, \quad NoSym(\text{Ben})@(181, 242]$$

If we want to know whether Ben is immune between September 8 and September 9, represented as a temporal fact $Ben@(251, 252]$, we can formulate this as a *fact entailment* problem: Is $Ben@(251, 252]$ entailed by $\mathcal{D}_{ex}$ and $\Pi_{ex}$?

Traditionally, a symbolic reasoner Bellomarini et al. (2018); Wang et al. (2022) is used to check entailment by applying temporal rules to temporal data, deriving new facts, and verifying if the given fact is among the derived ones. There are two key challenges in using symbolic reasoners for temporal reasoning tasks: 1) *symbolic reasoners cannot directly process natural language descriptions and instead require inputs to be formalized as logical rules*; 2) *generating these logically consistent and error-free rule representations is a non-trivial task that demands significant domain expertise and manual effort*.[2] In this paper, we explore whether LLMs can solve temporal reasoning tasks in both symbolic and natural language forms, potentially serving as an alternative to, or a complementary tool for, traditional symbolic reasoners. Our contributions are summarized as follows:

- `t-BEN` is the first temporal reasoning benchmark constructed based on the semantics of temporal logic, while supporting evaluation in both symbolic and natural language forms.

- `t-BEN` provides a scalable and verifiable testbed for the creation of datasets with varying sizes and rule structures of different complexities. Moreover, the questions in `t-BEN` are guaranteed to be unseen by LLMs during pretraining, thereby mitigating the risk of data leakage and enabling a more rigorous and trustworthy evaluation setting.

- We conduct extensive experiments to evaluate the performance of several frontier Large Language Models (LLMs), including both open-source and proprietary models, on `t-BEN`. Our results reveal an interesting observation: among all evaluated models, only DeepSeek-R1 delivers impressive results on `t-BEN`, while other LLMs—including GPT-4o—perform poorly, often nearing random chance. Additionally, our analysis of other distilled variants of DeepSeek-R1 reveals consistent performance gains, which we attribute to DeepSeek's unique training strategy—specifically, the inclusion of instruction-following data during the final stages of supervised fine-tuning and reinforcement learning training.

---

[1] If the arity is 0, then $P$ is treated as a statement that is either true or false. This differs from temporal knowledge graphs, which consist solely of quadruples (arity=2).

[2] Although prior work has explored converting natural language expressions into logical rules Chen et al. (2023); Tammet et al. (2024), the accuracy of such conversions remains an open question. The two-stage pipeline may suffer from error propagation, which complicates the reasoning process.

## 2 RELATED WORKS

**Temporal logic reasoning** Knowledge representation languages, such as Linear Temporal Logic (LTL) Huth & Ryan (2004) and DatalogMTL Brandt et al. (2018), have become the de facto standard for specifying temporal properties in both formal verification and artificial intelligence. Many temporal reasoning problems have proven to be PSPACE-complete Wałęga et al. (2019); Fionda & Greco (2018); Bauland et al. (2009). *Satisfiability checking*, that is, the problem of deciding whether a given formula admits a satisfying model, is one of the most important computational tasks associated with the logic, and one of the first that have been carefully studied Sistla & Clarke (1985). Similarly, the reasoning tasks considered in DatalogMTL are *fact entailment* and *consistency checking*. These problems polynomially reduce to the complements of each other Brandt et al. (2018). Despite this theoretically high computational complexity, numerous techniques and tools are developed to solve different temporal reasoning problems, ranging from tableau systems Goré & Widmann (2009); Bertello et al. (2016) to reductions to model checking Cavada et al. (2014), to automata techniques Li et al. (2014); Wang et al. (2022).

**Benchmarking and Reasoning in Large Language Models** Although the aforementioned temporal reasoning problems have been widely explored in the traditional logic-based AI domain, they remain underexplored in the regime of LLMs. In recent years, benchmarking reasoning capabilities in LLMs is a problem of pressing interest to the field Plaat et al. (2024); Chang et al. (2024); Huang & Chang (2023). There is a substantial body of research evaluating the reasoning abilities of LLMs, covering areas such as arithmetic reasoning, logical reasoning, and commonsense reasoning. Notably, simple math problem datasets like AQUA Ling et al. (2017), GSM8K (Cobbe et al., 2021), and SVAMP (Patel et al., 2021) are frequently used to assess arithmetic reasoning (Touvron et al., 2023; Shi et al., 2023). Welleck et al. (2021) developed NaturalProofs, a multi-domain dataset for studying mathematical reasoning in natural language, while Welleck et al. (2022) investigated LLMs' abilities to generate the next step in mathematical proofs and complete full proofs. Additionally, LLMs have been evaluated on logical reasoning tasks, including symbolic tasks like Coin Flip and Last Letter Concatenation (Wei et al., 2022), and Logic Grid Puzzles on the BIG-BENCH (Srivastava et al., 2023). Most relevant to our work are various approaches to evaluating and enhancing the algorithmic reasoning abilities of LLMs (Zhou et al., 2022; Fatemi et al., 2025).

## 3 DATALOGMTL

DatalogMTL Brandt et al. (2018); Wałęga et al. (2019) is a temporal logic language, which extends Datalog Abiteboul et al. (1995) with operators from metric temporal logic (MTL) Koymans (1990). Different Datalog designed to handle static facts and rules due to lack of built-in temporal constructs, DatalogMTL equipped with MTL operators is enabled to reasoning about properties of systems that evolve over time. These operators build upon the standard linear temporal logic (LTL) Huth & Ryan (2004) operators, such as $\diamondsuit$ standing for "sometime in the past", $\boxminus$ for "always in the past", and $\mathcal{S}$ for "since", as well as their future counterparts $\oplus$ for "sometime in the future", $\boxplus$ for "always in the future", and $\mathcal{U}$ for "until". In MTL, however, these LTL operators are annotated with intervals; for instance, the expression $\diamondsuit_{[1,2]} LiveIn(x,y)$ is true at time $t$ if entity $x$ lived in location $y$ sometime between times $t-1$ and $t-2$. Similarly, $\boxminus_{[1,2]} LiveIn(x,y)$ holds at time $t$ if $x$ continuously lived in $y$ throughout the aforementioned time interval.

**Syntax** We consider a *signature* consisting of pairwise disjoint countable sets of constants, variables, and predicates with non-negative integer arities. A term is either a constant or a variable. A *relational atom* is an expression of the form $P(\mathbf{s})$, with $P$ a predicate and $\mathbf{s}$ a tuple of terms whose length matches the arity of $P$. In this paper, we restrict ourselves to a fragment in which metric atoms are generated by the following grammar, where $P(\mathbf{s})$ is a relational atom and $\varrho$ an interval:

$$M ::= P(\mathbf{s}) \mid \diamondsuit_\varrho M \mid \oplus_\varrho M \mid \boxminus_\varrho M \mid \boxplus_\varrho M$$

A rule in this fragment is an expression of the form

$$P(\mathbf{s}) \leftarrow M_1 \wedge \cdots \wedge M_n, \qquad \text{for } n \geq 1, \tag{1}$$

where the body atoms $M_1, \ldots, M_n$ are metric atoms and the head atom $P(\mathbf{s})$ is relational. A program is a finite set of rules.

**Semantics**  An interpretation $\mathfrak{I}$ is a function assigning truth values to ground relational atoms $P(\mathbf{c})$ and time points $t \in \mathbb{Z}$. It determines if $P(\mathbf{c})$ is satisfied at $t$, denoted as $\mathfrak{I}, t \models P(\mathbf{c})$, or not, denoted as $\mathfrak{I}, t \not\models P(\mathbf{c})$. This notion of truth assignment extends to other ground metric atoms in the considered fragment as follows:

$$\mathfrak{I}, t \models \Diamondminus_\varrho M \qquad \text{iff} \qquad \mathfrak{I}, t' \models M \text{ for some } t' \text{ with } t - t' \in \varrho,$$
$$\mathfrak{I}, t \models \Diamondplus_\varrho M \qquad \text{iff} \qquad \mathfrak{I}, t' \models M \text{ for some } t' \text{ with } t' - t \in \varrho,$$
$$\mathfrak{I}, t \models \boxminus_\varrho M \qquad \text{iff} \qquad \mathfrak{I}, t' \models M \text{ for all } t' \text{ with } t - t' \in \varrho,$$
$$\mathfrak{I}, t \models \boxplus_\varrho M \qquad \text{iff} \qquad \mathfrak{I}, t' \models M \text{ for all } t' \text{ with } t' - t \in \varrho.$$

For example, an interpretation making atom $LiveIn(Ann, Paris)$ true everywhere within $[10, 30]$ and false elsewhere makes $\boxminus_{[1,2]} LiveIn(Ann, Paris)$ true at the time point 31, but false at 32. An interpretation can be alternatively seen as the (possibly infinite) set of facts that it satisfies, which yields a natural meaning to containment and minimality of interpretations.

## 3.1 Major Temporal Reasoning Problems

According to Brandt et al. (2018); Wałęga et al. (2019), temporal logic reasoning involves two major problems: *consistency checking* and *fact entailment*. *Consistency checking* is the task of determining whether a given program and dataset admit a common model Emerson (1990); Schnoebelen (2002). *Fact entailment* involves checking whether a program and dataset together entail a specific relational fact. Brandt et al. (2018) note that in DatalogMTL, consistency checking and fact entailment are complementary problems. Consequently, this paper focuses solely on the *fact entailment* problem to evaluate the temporal reasoning capabilities of large language models.

## 4  T-BEN: A Suite for Generating Temporal Reasoning Datasets

DatalogMTL is a temporal logic language that can characterize complex temporal conditions by defining various rules using combinations of different atoms and temporal operators ($\Diamondminus, \Diamondplus, \boxminus, \boxplus$) whose semantics has been described in Section 3. To some extent, the complexity of a *fact entailment* problem is largely determined by the complexity of associated temporal rules.

---

**Zero-shot Prompt Prefix**

Given a dataset, temporal rules and a temporal fact, you need to apply the rules to the dataset and then judge whether the given fact is entailed by the dataset and rules.

The rules are expressed as DatalogMTL, a language of temporal logic that extends Datalog with operators from metric temporal logic (MTL). The semantics of four MTL operators are given as follows:

If $\Diamondminus_{[a,b]} A$ is true at the time t, it requires that A needs to be true at some time between t-b and t-a.

If $\boxminus_{[a,b]} A$ is true at the time t, it requires that A needs to be true continuously between t-b and t-a.

If $\Diamondplus[a, b] A$ is true at the time t, it requires that A needs to be true at some point between t+a and t+b.

If $\boxplus[a, b] A$ is true at the time t, it requires that A needs to be true continuously between t+a and t+b.

Now, we have a data, some DatalogMTL rules and a fact entailment question. You should only output true or false, and please do not output other words.

---

**SingleAtom**

**R**: $A \leftarrow \Diamondminus_{[1,2]} B$
**D**: $\{B@[4, 5]\}$
**Q**: A@6 is entailed?

**MultiAtoms**

**R**: $A \leftarrow \boxminus_{[3]} B \land \boxminus_{[2, 3]} C$
**D**: $\{B@[1], C@[1, 3]\}$
**Q**: A@3 is entailed?

**Rational**

**R**: $A \leftarrow \boxminus_{[1.2, 2.1]} B \land \boxminus_{[4.2, 5.1]} B$
**D**: $\{A@[1.1]\}$
**Q**: A@[2.4, 2.8]) is entailed?

**MixedOperators**

**R**: $A \leftarrow \Diamondminus_{[1, 2.4]} B \land \boxplus_{[1, 2]} C$
**D**: $\{B@[1], C@[2, 4]\}$
**Q**: A@2.3 is entailed?

**MultiRules**

**R**: $D \leftarrow \boxminus_{[2]} B \land \boxplus_{[1,2]} C$
        $A \leftarrow \Diamondminus_{[1.5, 2]} D \land \Diamondminus_{[2]} C$
**D**: $\{B@[1], C@[2, 5]\}$
**Q**: A@[4.5, 5] is entailed ?

**Recursive**

**R**: $A \leftarrow \Diamondminus_{[1,2]} A \land \boxminus_{[1,10]} C$
**D**: $\{A@[1], C@[1, 100]\}$
**Q**: A@99 is entailed?

Figure 1: Six levels of temporal reasoning problems. We present an example representing each level, along with the corresponding rule, dataset, and fact entailment problem. A zero-shot-prompt prefix is provided. For better demonstration, we use the symbols $\Diamondminus, \Diamondplus, \boxminus,$ and $\boxplus$, which are replaced by $< ->, < +>, [-],$ and $[+]$, respectively, in the actual prompts due to typing constraints.

## 4.1 Leveling DatalogMTL Rules

To address the aforementioned challenge and provide a more comprehensive evaluation of the temporal reasoning abilities of large language models, we aim to create a new synthetic benchmark with flexible configurations for customizing rule structures and task complexity. We classify DatalogMTL rules into six classes (**S-Atom**, ..., **Recursive**) based on their structural representations, considering factors such as the number of body atoms, the number of temporal operators used, the number of rules involved, and whether the rules are recursive. While we are unable to quantify the degree of complexity of each level, we assume that higher levels correspond to greater complexity. This assumption is based on the observation that more complex rule structures require additional temporal reasoning steps when using a symbolic reasoner like MeTeoR Wang et al. (2022).

**S-Atom**    The most simplest form of a rule is $A \leftarrow \oslash_{[\rho]} B$, where $\oslash$ could be one of the four metric temporal operators ( $\boxminus$, $\boxplus$, $\Diamonddown$ and $\Diamondup$). We ensure that $A$ and $B$ are two different atoms, so only one calculation operation. A **S-Atom** example is given in Figure 1, where we can derive $A@[5, 7]$ based on the given dataset and the rule, entailing that $A@6$ is true. In particular, we consider the integer timeline, a fragment of DatalogMTL Wałęga et al. (2020) and use one type of MTL operator.

**M-Atoms**    In the **S-Atom**, the body contains only one atom, so a single rule application is sufficient to complete the derivation. In **M-Atoms**, we increase the number of atoms in the rule body, requiring not only the validation of each atom but also an intersection operation to obtain the final valid interval. As the example shown in Figure 1, the rule contains two atoms. First, we calculate the valid intervals for each atom. Based on the provided facts, $\boxminus_{[3]} B$ holds only at the punctual time point $[4, 4]$, and $\boxminus_{[2,3]} A$ holds at the interval $[4,5]$. The intersection of these intervals, $[4, 4]$ and $[4, 5]$, is $[4, 4]$. We derive that $A$ is true at the time point 4, so $A@4$ is entailed. As with **S-Atom**, we consider DatalogMTL over the integer timeline Wałęga et al. (2020) and use only one type of MTL operator.

**Rational**    Both **S-Atom** and **M-Atoms** focus solely on the integer timeline, which represents a relatively limited time space and simplifies reasoning due to the integer semantics Wałęga et al. (2020). In **Rational**, we build on top of **M-Atoms** by expanding the timeline to include the rational numbers, incorporating decimal time points. Intuitively, rational-based numerical operations are more complex than their integer-based counterparts, and we aim to determine if large language models exhibit similar behavior. We continue to use only one type of MTL operator at this level.

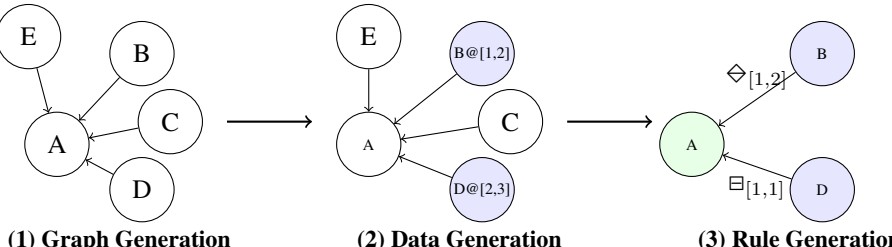

**(1) Graph Generation**          **(2) Data Generation**          **(3) Rule Generation**

Figure 2: An example of generating temporal data and rules. First, we randomly generate a graph. Next, our program selects specific nodes to assign time points. In our example, nodes B and D are chosen, resulting in two temporal facts: $\{B@[1,2], D@[2,3]\}$; Finally, we select a node as the head atom, with body atoms derived from the previous step. We then randomly assign temporal operators to these body atoms, resulting in the rule: $A \leftarrow \Diamonddown_{[1,2]} B \wedge \boxminus_{[1,1]} D$. The number of body atoms, the time range, and the temporal operators are specified as input parameters.

**M-Operators**    Using only one operator limits the expressiveness of DatalogMTL, preventing the definition of complex temporal conditions. Thus, a natural expansion is to allow the use of MTL operators. The four types of MTL operators can be used to define temporal conditions associated with both the past and the future. A **M-Operators** example is shown in Figure 1, which involves two MTL operators ($\Diamonddown$ and $\boxplus$). To complete the derivation, we first calculate the valid interval where $\Diamonddown_{[1,2.4]} B$ with the past operator ($\Diamonddown$) holds, which is $[2, 3.4]$. Then, we calculate $\boxplus_{[1,2]} C$, whose valid

interval is $[1, 2]$. After performing the interval intersection, we obtain that A holds at the time interval $[2, 2]$. Thus, the temporal $A@2.3$ is not entailed.

**M-Rules**  In the previous four levels, fact entailment is associated with only one temporal rule. However, in more practical scenarios, multiple temporal rules may be required to express complex temporal conditions. In this level, we consider a multi-rule temporal reasoning case, where fact entailment involves multiple temporal rules and rule applications must be executed across these rules to complete the derivation. As the example in Figure 1, to derive the target atom $A$, we need to know both $D$ and $C$. However,the dataset only provides the information about $C$. We can derive the $D$ holds at 3 according to the first temporal rule $D \leftarrow \boxminus_{[2]} \wedge \boxplus_{[1,2]} C$; then, we can derive that $A$ holds at the interval $[4.5, 5]$ according to the second rule. Hence, $A@[4.5, 5]$ is entailed.

**Recursive**  The fact entailment problem at this level is considered the hardest because it involves recursion. Unlike static knowledge representation languages (e.g., Datalog), where all facts can be derived after a certain number of rule applications, some recursive rules in DatalogMTL may require an infinite number of applications. Even for symbolic-based approaches, this presents a significant challenge, and researchers have devoted considerable effort to addressing it Wałęga et al. (2021; 2023). According to Wałęga et al. (2023), in the recursive scenarios, periodic structures will ultimately occur repeatedly, but calculating these periodic structures is challenging. From a human perspective, however, identifying such periodic structures can be straightforward. For instance, consider a recursive rule $\boxplus_{1year} \text{Bday}(x) \leftarrow \text{Bday}(x)$, which states that anyone having their birthday at a time point t will also be having their birthday at the same time the following year. If we know that Ben has his birthday on Jun 8, 1991, it is easy to know that he will have his birthday on Jun 8, 1992, Jun 8, 1993 and so on. However, this is difficult for traditional symbolic-based approaches to handle. Therefore, we design fact entailment problems associated with recursive rules to test whether large language models can perform well in this setting.

Specifically, we use facts from both propositional logic Klement (2004) and first-order logic Barwise (1977). The former contains declarative statements that are either 'true' or 'false', while the latter includes expressions with one or more variables. For example, we allow both forms of temporal facts: $Raining$ and $Immune(x)$. The former states that an event (raining) is occurring, while the latter denotes that a property (immune) is associated with an entity, where $x$ acts as a placeholder that can be instantiated to any entity, such as $Immune(Ben)$, indicating that Ben is immune.

## 4.2 GENERATING TEMPORAL DATA AND RULES

The benchmark generation process[3] can be mainly divided into the following three steps: 1) Graph construction, 2) Data generation, and 3) Rule generation.

**Graph construction**  We employ a general-purpose random graph generator to generate a connected directed random graph. The nodes in the random graph represent predicates, such as A, B, and C. Each edge in this graph represents a body atom of a rule pointing to the corresponding head in the rule. In particular, a predicate can appear in bodies of multiple different rules.

**Data generation**  After the construction of the graph, the program will traverse each nodes in the graph and randomly assign time points or time intervals to the chosen nodes. The time points or intervals are generated based on a given range.

**Rule Generation**  Once the temporal data is generated, the rule generator traverses the edges of the graph, assigning random operators and intervals to the edges. To ensure the generated graph is non-trivial, a reasoning process is performed across the entire graph after completing this step to ensure new facts can be inferred. If multiple rules are required, the program repeats previous steps until a sufficient number of rules are generated.

**An example**  Figure 2 shows an example of generating temporal data and rules. In particular, our program will have a post-processing operation to scan all the data and rules to ensure they have

---

[3]The pseudocode for this benchmark generation algorithm can be found in Appendix F.

been utilized and removes any data and rules that are not participated in the the temporal reasoning process. We define the following flags for the samples to be generated based on their characteristics: `rational number`, `multiple body atoms`, `recursive` and `mixed operators`. These flags control the rule structures during the generation process.

| | Prompt type | S-Atom | M-Atoms | Rational | M-Operators | M-Rules | Recursive |
|---|---|---|---|---|---|---|---|
| o3 | Zero-shot | **100.0** | **100.0** | 99.5 | **100.0** | 99.5 | **98.5** |
| gemini-2.5-pro | Zero-shot | 96.0 | 92.0 | 98.5 | 94.0 | 94.0 | 99.0 |
| gemini-2.5-flash | Zero-shot | 96.5 | 91.0 | 98.0 | 92.0 | 94.5 | 94.5 |
| gemini-2.5-flash-lite | Zero-shot | 46.0 | 50.0 | 49.5 | 50.5 | 50.5 | 39.5 |
| | Zero-shot-CoT | 97.5 | 88.5 | 91.5 | 96.0 | 91.5 | 61.0 |
| Claude 3.5v2 | Zero-shot | 61.0 | 58.0 | 58.0 | 66.5 | 51.0 | 49.5 |
| | Zero-shot-CoT | 86.0 | 75.0 | 84.5 | 89.0 | 75.0 | 56.0 |
| GPT-4o | Zero-shot | 45.8 | 43.2 | 37.1 | 57.3 | 53.3 | 37.7 |
| | Few-shot | 40.4 | 38.0 | 27.2 | 51.6 | 36.7 | 32.2 |
| | Zero-shot-CoT | 85.6 | 85.1 | 85.7 | 90.3 | 74.0 | 58.0 |
| Llama-3-8B | Zero-shot | 40.7 | 44.0 | 43.9 | 60.5 | 39.1 | 8.7 |
| | Few-shot | 38.4 | 44.3 | 44.4 | 47.1 | 36.1 | 30.2 |
| | Zero-shot-CoT | 59.9 | 58.4 | 68.2 | 64.1 | 59.0 | 48.5 |
| Qwen2.5-32B | Zero-shot | 47.0 | 46.0 | 33.0 | 49.5 | 38.5 | 16.0 |
| | Few-shot | 41.5 | 48.0 | 31.0 | 56.0 | 42.5 | 21.5 |
| | Zero-shot-CoT | 80.0 | 80.0 | 78.4 | 89.0 | 61.6 | 51.5 |
| Qwen3-32B | Zero-shot | 99.5 | 99.5 | 98.5 | 98.5 | 96.5 | 77.5 |
| DeepSeek-R1 | Zero-shot | **100.0** | 96.0 | 99.5 | **99.5** | 97.5 | 88.9 |
| Distill-Qwen-7B | Zero-shot | 80.7 | 75.9 | 70.0 | 79.9 | 65.6 | 45.5 |
| Distill-Qwen-14B | Zero-shot | 95.0 | 92.0 | 97.0 | 95.5 | 88.4 | 57.6 |
| Distill-Qwen-32B | Zero-shot | 96.9 | 87.9 | 97.5 | 90.4 | 86.2 | 64.0 |

Table 1: Accuracy of 13 models on our synthetic benchmarks across six rule structures.

## 5 EXPERIMENTS AND RESULTS

**Baselines** We evaluate the performance of seven LLMs on `t-BEN`. These models include proprietary models like GPT-4o Achiam et al. (2023), OpenAI-o3 OpenAI (2025), Gemini 2.5 series Comanici et al. (2025) and Claude 3.5Anthropic (2024). Open source models that we evaluate include Llama-3 Dubey et al. (2024) and Qwen2.5-32B-Instruct Yang et al. (2024), Qwen3-32B Yang et al. (2025), DeepSeek-R1 Liu et al. (2024) and three DeepSeek-R1 distilled models (DS-R1-Distill-Qwen-7B, DS-R1-Distill-Qwen-14B and DS-R1-Distill-Qwen-32B). We conduct experiments on all non-reasoning models using three different prompting strategies: zero-shot prompting, few-shot in-context learning (Brown et al., 2020), and chain-of-thought prompting (Wei et al., 2022). We consider only the zero-shot prompting setting for reasoning models due to the unique nature.

**Benchmark statistics and experimental settings** Unless otherwise specified, each benchmark level contains 200 samples selected from the facts derived using the chosen data and rule(s). For negative samples, a random interval is chosen, ensuring that these intervals do not overlap with those of the derived facts. For all baselines, the temperature value is set to 0. For few-shot prompting techniques, the input prompt includes two manually constructed exemplars. In this paper, we use both the F1 score and the accuracy as the evaluation metric. Single-run results are reported.

### 5.1 MAIN RESULTS

From Table 1, we observe a striking phenomenon: compared to DeepSeek-R1 and its distilled models, GPT-4o, Llama-3, and Qwen-32B-Instruct perform poorly on the temporal logic reasoning problems of `t-BEN`, even with chain-of-thought prompting (CoT). This suggests that these models lack the advanced reasoning capabilities necessary for truly understanding symbolic representations involving time. We also find that frontier reasoning models—including o3, Gemini 2.5 Pro, and Gem-

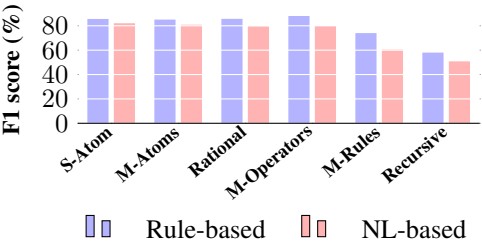

Figure 3: Comparison of symbolic and natural language (NL) based temporal logic reasoning using GPT-4o with Zero-shot-CoT.

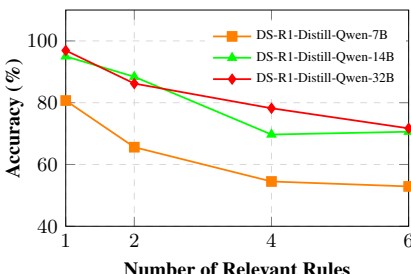

Figure 4: Performance of DeepSeek-R1 distilled models across different numbers of relevant rules.

ini 2.5 Flash—deliver strong results, with o3 approaching 100% accuracy across the six rule structures. Notably, in the **M-Rules** and **Recursive** benchmarks, we observe a significant performance drop for most of evaluated models. These results indicate that recursive rules pose a particular challenge, as they require not only an understanding of language semantics and step-by-step reasoning but also strong *inductive abilities*. However, a surprising finding is that, apart from the task involving recursive structures, DeepSeek-R1 achieves an accuracy of 88.9%, and for all five other levels, it surpasses 96% accuracy—demonstrating exceptionally strong symbolic reasoning abilities. One possible explanation for DeepSeek's strong performance lies in its distinctive training strategy—namely, the incorporation of instruction-following data during the final stages of supervised fine-tuning and reinforcement learning. This approach may improve the model's ability to adhere to prompts, such as our system-provided instructions, thereby enhancing its temporal reasoning capabilities. Besides, we evaluated several smaller DeepSeek-R1 distilled models, which also exhibited remarkable performance. These findings suggest that integrating instruction-following data into the training process may be an effective strategy for strengthening a model's temporal reasoning abilities.

## 5.2 SYMBOLIC V.S. NATURAL LANGUAGE

In addition to evaluating the temporal reasoning capabilities of LLMs in symbolic forms—where traditional symbolic reasoners excel—it is also valuable to assess their performance in natural language scenarios, which symbolic reasoners cannot handle. To this end, we adopt a common strategy of verbalizing logical rules before presenting them to the LLMs, following the approach explored in prior works Saxena et al. (2021); Ismayilzada et al. (2023). Given that manually converting each rule into its corresponding natural language expression is a labor-intensive process, we adopt a template-based approach to automate this verbalization. Although this method may result in unnatural expressions, it provides a practical alternative to manual translation.

From Figure 3, we observe that both the rule-based and natural language-based settings achieve similar results, with the rule-based approach performing slightly better. The comparison indicates that *LLMs are also capable of understanding the semantics of input expressed in rules*, provided that each notation is clearly explained in the instructions. Notably, both settings struggle with the **M-Rules** and **Recursive** cases. One possible reason for this is that, while LLMs can understand the semantics of temporal logic language, they still face significant challenges in executing multiple deductions, retaining intermediate results, and recognizing repeated patterns.

## 5.3 ABLATION STUDY

To explore which component of the rule structure most significantly impact the reasoning complexity for LLMs, we designed four sets of ablation study experiments using GPT-4o. These experiments explored the effects of the number of relevant rules , the number of operators considered, the percentage of irrelevant data, and the percentage of irrelevant rules. From Figure 5 (a), we observe that as the lengths of dependent rules increase, the model's performance noticeably degrades.One possible reason is that when multiple rules are mutually dependent, the model needs to store intermediate results during the derivation process to complete subsequent steps that rely on previously derived outcomes. Unlike symbolic reasoners, which can explicitly store intermediate results, it may

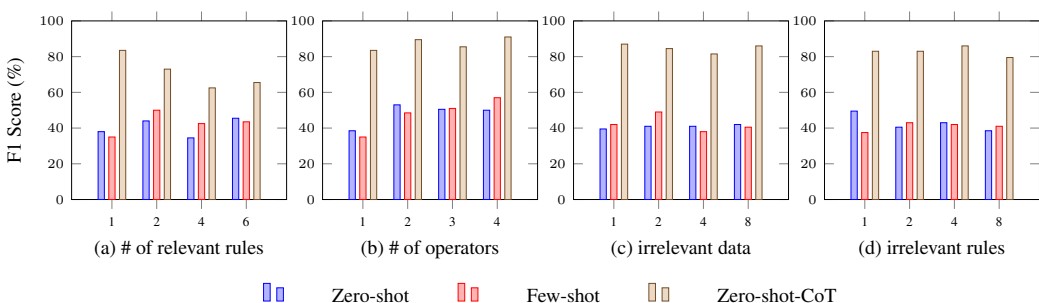

Figure 5: Results of ablation study for GPT-4o with three different prompting strategies.

be challenging for large language models (LLMs) to retain such information in an auto-regressive manner. Figure 5 (b) demonstrates that using more types of operators does not affect reasoning complexity, indicating that understanding the semantics of the temporal logic language is not a major issue for the model. Results in Figure 5 (c) and (d) show that the model's performance is minimally affected by irrelevant information, demonstrating its ability to correctly select relevant rules and remain resistant to distracting information.

In Figure 5, we observe that the number of relevant rules has the most significant impact. We experiment with the three DeepSeek-R1 distilled models, which have demonstrated strong performance in the single-rule setting (Table 1). In Figure 4 , it shows that as the number of relevant rules increases, performance declines, suggesting that reasoning over multiple rules remains a significant challenge.

**Robustness to the input formats** We investigate the impact of the input formats to the LLM-based approach via three evaluation settings: ① *error-free symbolic input*, ② *symbolic input with errors*, and ③ *natural language input*. We construct a subset of 100 questions, each represented in all three formats. For ③, we introduce syntactic errors by randomly removing notation elements that cause parsing issues—for example, altering $\Box_{[1,2]}$ to $\Box_{1,2]}$ by removing the opening bracket. Symbolic reasoners can only handle the error-free symbolic input. In contrast, the LLM demonstrates strong accuracy across all three settings (95.0%, 94.5% and 94.4%). This suggests that the LLM not only exhibits effective temporal reasoning capabilities but also shows robustness to imperfect input.

**Analysis of errors** We do a manual analysis of the reasoning processes of two models of the same size—Qwen2.5-32B-Instruct and DeepSeek-R1-Distill-Qwen-32B—in the most challenging recursive setting, we observe a key difference. Qwen2.5-32B-Instruct performs only shallow inference step, failing to recognize the recursive nature of the problem and its potential for infinite expansion. In contrast, DeepSeek-R1-Distill-Qwen-32B correctly identifies the recursive structure, explicitly acknowledging it with statements such as "... applying the rule again, A at 8 would imply A at 10, and so on." This deeper understanding enables the model to arrive at the correct result.

In addition, manually inspecting the CoT reasoning for each failure case to identify the precise source of error is challenging. Hence, to establish a systematic taxonomy of failure types and gain insights into the diverse reasoning behaviors of LLMs, we devised an automated method that leverages the Gemini-2.5-Pro model as a proxy to analyze CoT traces from failure cases generated by Qwen-2.5. We identified six recurring categories of errors: 1) failure to apply rules recursively; 2) incorrect interval overlap or boundary checks; 3) misinterpretation of query semantics; 4) incorrect interval calculation; 5) misinterpretation of operator semantics, and 6) other logical or factual errors.

## 6 CONCLUSION

We introduce `t-BEN`, a benchmark suite designed to systematically evaluate the temporal reasoning capabilities of large language models (LLMs) in a controlled setting. Preliminary results suggest that certain LLMs, such as DeepSeek-R1, may serve as viable alternatives or complementary tools to traditional symbolic reasoners, though further investigation is needed. By open-sourcing our codes and datasets, we hope to stimulate further research and development in this field, thereby better facilitating the potential application of LLMs in traditional logic-based AI domains.

## 7 REPRODUCIBILITY STATEMENT

This paper can be reproduced through the released dataset and scripts in the supplemental material. Proprietary models are available through their respective vendors, and open source models can be retrieved through HuggingFace.

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

# Appendix

## A    DATASHEETS FOR DATASETS

To help the community better understand the dataset, we present the datasheets of the `t-BEN` dataset, according to Gebru et al. (2021).

### A.1    MOTIVATION

**Purpose**    The dataset is used as a benchmark to test the LLM's reasoning ability on temporal logic. Temporal logic reasoning involves both logic reasoning and numerical reasoning, and the ability is useful in many downstream tasks. The benchmark specifically addressed the bias issue caused by data leakage by generating data randomly and automatically. Since it can be scaled up easily, it might also be used to fine tune a model to enhance its reasoning abilities.

**Creators / Funding**    Those information will be disclosed once the paper is accepted.

### A.2    COMPOSITION

**Instance**    All instances in the dataset are a temporal reasoning question written in DatalogMTL.

| Type of Sample | # of Positive Samples | # of Negative Samples |
|---|---|---|
| SingleAtom | 500 | 500 |
| MultiAtoms | 300 | 300 |
| Rational | 500 | 500 |
| MixedOperators (with 2 operators) | 1739 | 1739 |
| MixedOperators (with 3 operators) | 145 | 145 |
| MixedOperators (with 3 operators) | 126 | 126 |
| MultiRules (with 2 rules) | 250 | 250 |
| MultiRules (with 4 rules) | 250 | 250 |
| MultiRules (with 6 rules) | 150 | 150 |
| Recursive | 500 | 500 |

Table 2: The number of samples of different categories in our dataset

**Size**    Depending on the complexity of the reasoning problems, we divided the dataset into six sub dataset, the number of instances are listed in Table 2.

For MultiAtoms, we don't specify the number of operators it has in the rule nor evaluate them separately, while in general it follows the following distribution presented in Table 3.

Note that the dataset doesn't contain all possible instances. There are infinite number of possible instances.

**Instance Details**    Each instance contains a data field, which is a set of the known variables, a set of rules, a single query and a boolean value indicating that if the query is true. They are represented in JSON format.

| Type of Sample | # of Positive Samples | # of Negative Samples |
|---|---|---|
| MultiAtoms (with 2 atoms in the rule) | 109 | 115 |
| MultiAtoms (with 3 atoms in the rule) | 79 | 79 |
| MultiAtoms (with 4 atoms in the rule) | 61 | 64 |
| MultiAtoms (with 5 atoms in the rule) | 51 | 42 |
| Total | 300 | 300 |

Table 3: The distribution of the number of atoms in our MultiAtoms subset of our dataset

**Label**  Yes, the label is presented for each instace in the dataset.

**Missing Information**  No, all information is completed.

**Relationships**  All instances are independent in our dataset.

**Splits**  There isn't a recommended data split for our dataset.

**Errors**  No, there isn't any error in our dataset. All instances are verified to be correct.

**Self-contained**  Yes, the dataset is self-contained, no external resource is required.

**Confidentiality**  No, all data is considered as public.

### A.3  COLLECTION PROCESS

The dataset is generated automatically without input from the real world. The generation algorithm is presented in Section 4

### A.4  PROCESSING

We used The Metric Temporal Reasoner (MeTeoR) to verify all generated instances.

### A.5  USE

The dataset is intended to be used as a metric to evaluate the general LLM's temporal reasoning ability.

### A.6  DISTRIBUTION

The dataset will be publicly available on HuggingFace with no restrictions on re-distribution upon acceptance.

### A.7  MAINTENANCE

The dataset will be hosted on HuggingFace platform and the contact information of the dataset creator will be released upon acceptance.

## B   PROMPTS USED IN THE BASIC EVALUATION

For all evaluations, we prepend a system message to introduce the syntax of DatalogMTL language as below:

> You are given a dataset and a temporal rule, and your task is to judge whether the given fact is entailed by the dataset and the rule.
> The rules are expressed as DatalogMTL, a knowledge representation language that extends Datalog with operators from metric temporal logic (MTL). The semantics of four MTL operators are given as follows:
> If Diamondminus[a,b]A is true at the time t, it requires that A needs to be true at some time between t-b and t-a.
> If Boxminus[a,b]A is true at the time t, it requires that A needs to be true continuously between t-b and t-a.
> If Diamondplus[a,b]A is true at the time t, it requires that A needs to be true at some point between t+a and t+b.
> If Boxplus[a,b]A is true at the time t, it requires that A needs to be true continuously between t+a and t+b.

**Zero-shot**   For zero-shot evaluations, as well as all DeepSeek evaluations, the system prompt we uses is the above general introduction plus the statement: *You should not give any explanation and you should only output "true" or "false"*. We are using the statement *Now we have some temporal data and some rules, data: {data} rule: {rule}, Is {inquiry} true or not?* as the user prompt to evaluate LLM's reasoning ability.

Here is an example of the complete prompt we constructed to do zero-shot evaluation.

| System Prompt | You are given a dataset and a temporal rule, and your task is to judge whether the given fact is entailed by the dataset and the rule. The rules are expressed as DatalogMTL, a knowledge representation language that extends Datalog with operators from metric temporal logic (MTL). The semantics of four MTL operators are given as follows: If Diamondminus[a,b]A is true at the time t, it requires that A needs to be true at some time between t-b and t-a. If Boxminus[a,b]A is true at the time t, it requires that A needs to be true continuously between t-b and t-a. If Diamondplus[a,b]A is true at the time t, it requires that A needs to be true at some point between t+a and t+b. If Boxplus[a,b]A is true at the time t, it requires that A needs to be true continuously between t+a and t+b. You should not give any explanation and you should only output "true" or "false" |
|---|---|
| User Prompt | Now we have some temporal data and some rules, data: B@[3,10] rule: A:-Diamondplus[6,10]B Is A@[1,4] true or not? |
| LLM's output | false |
| Expected Answer | true |

**Few-shot**   For few-shot evaluations, just like the zero-shot case, the system prompt we uses is the above general introduction plus the statement: *You should not give any explanation and you should only output "true" or "false"*. However, in the user prompt, we are integrating some examples using the following syntax:

> To help you better understand the task, I will provide two examples.
> Example 1: data: {pos data} rule: {pos rule} in this case you should output "true" for {pos inquiry}.
> Example 2: data: {neg data} rule: {neg rule} in this case you should output "false" for {neg inquiry}.
> Now we have some temporal data and some rules, data: {data} rule: {rule}
> Is {inquiry} true or not?"

*{pos data}*, *{pos rule}* and *{pos inquiry}* are from a positive sample, *{neg data}*, *{neg rule}* and *{neg inquiry}* are from a negative sample. They are samples not in the testing set, but has the same type as the testing samples.

Here is an example of the complete prompt we constructed to do few-shot evaluation.

| System Prompt | You are given a dataset and a temporal rule, and your task is to judge whether the given fact is entailed by the dataset and the rule. |
|---|---|
| | The rules are expressed as DatalogMTL, a knowledge representation language that extends Datalog with operators from metric temporal logic (MTL). The semantics of four MTL operators are given as follows: |
| | If Diamondminus[a,b]A is true at the time t, it requires that A needs to be true at some time between t-b and t-a. |
| | If Boxminus[a,b]A is true at the time t, it requires that A needs to be true continuously between t-b and t-a. |
| | If Diamondplus[a,b]A is true at the time t, it requires that A needs to be true at some point between t+a and t+b. |
| | If Boxplus[a,b]A is true at the time t, it requires that A needs to be true continuously between t+a and t+b. |
| | You should not give any explanation and you should only output "true" or "false" |
| User Prompt | To help you better understand the task, I will provide two examples. |
| | Example 1: data: B@[5,7] |
| | rule: A:-Boxminus[10,12]B |
| | in this case you should output "true" for A@[17,17] |
| | Example 2: data: B@[1,9] |
| | rule: A:-Diamondplus[3,3]B |
| | in this case you should output "false" for A@[-25,-6] |
| | Now we have some temporal data and some rules, data: B@[3,10] |
| | rule: A:-Diamondplus[6,10]B |
| | Is A@[1,4] true or not?" |
| LLM's output | false |
| Expected Answer | true |

**Zero-shot-CoT** For zero-shot-cot evaluations, the system prompt we uses is the above general introduction without the additional the statement that we used in zero-shot or few-shot to ask LLM's to answer directly. We are going to put some directions to let LLM think step by step in the user prompt below:

> Now we have some temporal data and some rules, data: {data}
> rule: {rule}
> Is {inquiry} true or not? Do not answer directly, think step by step.

After that, we will put the LLM's response in the context and then asks LLM again about the final answer by *Based on your previous response, without any explanation state your answers to the question. You should only output "true" or "false".*

Here is an example of the complete prompt we constructed to do zero-shot evaluation.

| System Prompt | You are given a dataset and a temporal rule, and your task is to judge whether the given fact is entailed by the dataset and the rule. The rules are expressed as DatalogMTL, a knowledge representation language that extends Datalog with operators from metric temporal logic (MTL). The semantics of four MTL operators are given as follows: If Diamondminus[a,b]A is true at the time t, it requires that A needs to be true at some time between t-b and t-a. If Boxminus[a,b]A is true at the time t, it requires that A needs to be true continuously between t-b and t-a. If Diamondplus[a,b]A is true at the time t, it requires that A needs to be true at some point between t+a and t+b. If Boxplus[a,b]A is true at the time t, it requires that A needs to be true continuously between t+a and t+b. |
|---|---|
| User Prompt | Now we have some temporal data and some rules, data: B@[3,10] rule: A:-Diamondplus[6,10]B Is A@[1,4] true or not? Do not answer directly, think step by step. |
| LLM's output | First, let's understand the rule: A:-Diamondplus[6,10]B. This rule means that if B is true at some point between t+6 and t+10, then A is true at time t. Now, let's look at the data: B@[3,10]. This means that B is true at some point between time 3 and time 10. We are asked to determine if A@[1,4] is true or not. This means we are looking to see if A is true at some point between time 1 and time 4. Given the rule, for A to be true at time 1, B would need to be true at some point between time 7 (1+6) and 11 (1+10). For A to be true at time 4, B would need to be true at some point between time 10 (4+6) and 14 (4+10). However, B is only true between time 3 and time 10. Therefore, B is not true at any point between time 7 and 11 or between time 10 and 14. So, A@[1,4] is not true. |
| User Prompt | Based on your previous response, without any explanation state your answers to the question. You should only output "true" or "false" |
| LLM's output | false |
| Expected Answer | true |

## C  PROMPTS USED IN THE NATURAL LANGUAGE EVALUATION

For the evaluations using natural language, we no longer need to introduce the syntax of DatalogMTL language. Therefore there is no general system prompt.

**Zero-shot**   For zero-shot setting, we change the system prompt to a simple sentence to ensure that LLMs output in a desired format *You should not give any explanation and you should only output "true" or "false"*. We are using the statement *Now we have some temporal data and some rules, data: {data} rule: {rule}, Is {inquiry} true or not?* as the user prompt to evaluate LLM's reasoning ability. {data}, {rule} and {inquiry} are all replaced by their verbalized representation.

Here is an example of the complete prompt we constructed to do zero-shot evaluation.

| System Prompt | You should not give any explanation and you should only output "true" or "false" |
|---|---|
| User Prompt | Now we have some temporal data and some rules, data: A holds From 10.000 to 10.000 rule: B holds in each time such that A will hold sometime between 4.000 and 15.000 hours in the future Is B holds From -5.000 to 1.000 true or not? |
| LLM's output | false |
| Expected Answer | true |

**Few-shot**   For few-shot evaluations, just like the zero-shot case, the system prompt we uses is the same: *You should not give any explanation and you should only output "true" or "false"*. However, in the user prompt, we are integrating some examples using the following syntax:

---

To help you better understand the task, I will provide two examples.
Example 1: data: {pos data} rule: {pos rule} in this case you should output "true" for {pos inquiry}.
Example 2: data: {neg data} rule: {neg rule} in this case you should output "false" for {neg inquiry}.
Now we have some temporal data and some rules, data: {data} rule: {rule}

---

*{pos data}*, *{pos rule}* and *{pos inquiry}* are verbalized representations from a positive sample, *{neg data}*, *{neg rule}* and *{neg inquiry}* are verbalized representations from a negative sample. They are samples not in the testing set, but has the same type as the testing samples.

Here is an example of the complete prompt we constructed to do few-shot evaluation.

| System Prompt | You should not give any explanation and you should only output "true" or "false" |
|---|---|
| User Prompt | To help you better understand the task, I will provide two examples. |
| | Example 1: data: |
| | B holds From 5.000 to 7.000 |
| | rule: A holds in each time such that B did hold continuously for 10.000 and 12.000 hours before |
| | in this case you should output "true" for A holds From 17.000 to 17.000 |
| | Example 2: data: |
| | B holds From 1.000 to 9.000 |
| | rule: A holds in each time such that B will hold sometime between 3.000 and 3.000 hours in the future |
| | in this case you should output "false" for A holds From -25.000 to -6.000 |
| | Now we have some temporal data and some rules, data: |
| | B holds From 3.000 to 10.000 |
| | rule: A holds in each time such that B will hold sometime between 6.000 and 10.000 hours in the future |
| | Is A holds From 1.000 to 4.000 true or not? |
| LLM's output | true |
| Expected Answer | true |

**Zero-shot-CoT**   For verbalized zero-shot-cot evaluations, we no longer use system message since we don't need to introduce the synatx nor ask it to answer in a specific format. We organize our user prompts in the following format:

---

Now we have some temporal data and some rules, data: {data}
rule: {rule}
Is {inquiry} true or not? Do not answer directly, think step by step.

---

where {data}, {rule} and {inquiry} refers to the verbalized rules.

After that, we will put the LLM's response in the context and then asks LLM again about the final answer by *Based on your previous response, without any explanation state your answers to the question. You should only output "true" or "false"*.

Here is an example of the complete verbalized prompt we constructed to do zero-shot evaluation.

| System Prompt | |
|---|---|
| User Prompt | Now we have some temporal data and some rules, data: 
 B holds From 3.000 to 10.000 
 rule: A holds in each time such that B will hold sometime between 6.000 and 10.000 hours in the future 
 Is A holds From 1.000 to 4.000 true or not? Do not answer directly, think step by step. |
| LLM's output | To determine if A holds from 1.000 to 4.000 based on the given rule and data, let's break it down step by step: 
 1. **Understand the Data:** 
 - B holds from 3.000 to 10.000. 
 2. **Understand the Rule:** 
 - A holds at each time such that B will hold sometime between 6.000 and 10.000 hours in the future. 
 3. **Analyze the Time Intervals:** 
 - We need to check if B holds in the interval [6.000, 10.000] hours in the future from any time within [1.000, 4.000]. 
 4. **Check Each Time Point in [1.000, 4.000]:** 
 - For time 1.000: 
 - 6.000 to 10.000 hours in the future from 1.000 is [7.000, 11.000]. 
 - B holds from 3.000 to 10.000, which overlaps with [7.000, 10.000]. 
 - Therefore, B holds sometime in [7.000, 11.000]. 
 - For time 2.000: 
 - 6.000 to 10.000 hours in the future from 2.000 is [8.000, 12.000]. 
 - B holds from 3.000 to 10.000, which overlaps with [8.000, 10.000]. 
 - Therefore, B holds sometime in [8.000, 12.000]. 
 - For time 3.000: 
 - 6.000 to 10.000 hours in the future from 3.000 is [9.000, 13.000]. 
 - B holds from 3.000 to 10.000, which overlaps with [9.000, 10.000]. 
 - Therefore, B holds sometime in [9.000, 13.000]. 
 - For time 4.000: 
 - 6.000 to 10.000 hours in the future from 4.000 is [10.000, 14.000]. 
 - B holds from 3.000 to 10.000, which overlaps with [10.000, 10.000]. 
 - Therefore, B holds sometime in [10.000, 14.000]. 
 5. **Conclusion:** 
 - For each time point in the interval [1.000, 4.000], B holds sometime in the future interval [6.000, 10.000] hours from that time point. 
 - Therefore, according to the rule, A holds from 1.000 to 4.000. 
 So, based on the given rule and data, A holds from 1.000 to 4.000. |
| User Prompt | Based on your previous response, without any explanation state your answers to the question. You should only output "true" or "false" |
| LLM's output | true |
| Expected Answer | true |

## D  BAD CASE ANALYSIS

### D.1  CASES THAT ARE INCORRECT IN SYMBOLIZED REPRESENTATION BUT ARE CORRECT IN VERBALIZED REPRESENTATION

| | Symbolic | Verbalized |
|---|---|---|

| System Prompt | You are given a dataset and a temporal rule, and your task is to judge whether the given fact is entailed by the dataset and the rule. The rules are expressed as DatalogMTL, a knowledge representation language that extends Datalog with operators from metric temporal logic (MTL). The semantics of four MTL operators are given as follows: If Diamondminus[a,b]A is true at the time t, it requires that A needs to be true at some time between t-b and t-a. If Boxminus[a,b]A is true at the time t, it requires that A needs to be true continuously between t-b and t-a. If Diamondplus[a,b]A is true at the time t, it requires that A needs to be true at some point between t+a and t+b. If Boxplus[a,b]A is true at the time t, it requires that A needs to be true continuously between t+a and t+b. | (No introduction to the syntax is required for verbalized representations) |
|---|---|---|
| User Prompt | Now we have some temporal data and some rules, data: B@[3,10] rule: A:-Diamondplus[6,10]B Is A@[1,4] true or not? Do not answer directly, think step by step. | Now we have some temporal data and some rules, data: B holds From 3.000 to 10.000 rule: A holds in each time such that B will hold sometime between 6.000 and 10.000 hours in the future Is A holds From 1.000 to 4.000 true or not? Do not answer directly, think step by step. |

| LLM's output | First, let's understand the rule: A:-Diamondplus[6,10]B. This rule means that if B is true at some point between t+6 and t+10, then A is true at time t.

Now, let's look at the data: B@[3,10]. This means that B is true at some point between time 3 and time 10.

We are asked to determine if A@[1,4] is true or not. This means we are looking to see if A is true at some point between time 1 and time 4.

Given the rule, for A to be true at time 1, B would need to be true at some point between time 7 (1+6) and 11 (1+10). For A to be true at time 4, B would need to be true at some point between time 10 (4+6) and 14 (4+10).

However, B is only true between time 3 and time 10. Therefore, B is not true at any point between time 7 and 11 or between time 10 and 14. So, A@[1,4] is not true. | To determine if A holds from 1.000 to 4.000 based on the given rule and data, let's break it down step by step:
1. **Understand the Data:**
- B holds from 3.000 to 10.000.
2. **Understand the Rule:**
- A holds at each time such that B will hold sometime between 6.000 and 10.000 hours in the future.
3. **Analyze the Time Intervals:**
- We need to check if B holds in the interval [6.000, 10.000] hours in the future from any time within [1.000, 4.000].
4. **Check Each Time Point in [1.000, 4.000]:**
- For time 1.000:
- 6.000 to 10.000 hours in the future from 1.000 is [7.000, 11.000].
- B holds from 3.000 to 10.000, which overlaps with [7.000, 10.000].
- Therefore, B holds sometime in [7.000, 11.000].
- For time 2.000:
- 6.000 to 10.000 hours in the future from 2.000 is [8.000, 12.000].
- B holds from 3.000 to 10.000, which overlaps with [8.000, 10.000].
- Therefore, B holds sometime in [8.000, 12.000].
- For time 3.000:
- 6.000 to 10.000 hours in the future from 3.000 is [9.000, 13.000].
- B holds from 3.000 to 10.000, which overlaps with [9.000, 10.000].
- Therefore, B holds sometime in [9.000, 13.000].
- For time 4.000:
- 6.000 to 10.000 hours in the future from 4.000 is [10.000, 14.000].
- B holds from 3.000 to 10.000, which overlaps with [10.000, 10.000].
- Therefore, B holds sometime in [10.000, 14.000].
5. **Conclusion:**
- For each time point in the interval [1.000, 4.000], B holds sometime in the future interval [6.000, 10.000] hours from that time point.
- Therefore, according to the rule, A holds from 1.000 to 4.000.
So, based on the given rule and data, A holds from 1.000 to 4.000. |

| User Prompt | Based on your previous response, without any explanation state your answers to the question. You should only output "true" or "false" | |
|---|---|---|
| LLM's output | false | true |
| Expected Answer | true | |

## D.2 CASES THAT ARE INCORRECT IN ZERO-SHOT BY ARE CORRECT WHEN CHAIN OF THOUGHT IS APPLIED

Example 1:

| | Without CoT | With CoT |
|---|---|---|
| System Prompt | You are given a dataset and a temporal rule, and your task is to judge whether the given fact is entailed by the dataset and the rule.
The rules are expressed as DatalogMTL, a knowledge representation language that extends Datalog with operators from metric temporal logic (MTL). The semantics of four MTL operators are given as follows:
If Diamondminus[a,b]A is true at the time t, it requires that A needs to be true at some time between t-b and t-a.
If Boxminus[a,b]A is true at the time t, it requires that A needs to be true continuously between t-b and t-a.
If Diamondplus[a,b]A is true at the time t, it requires that A needs to be true at some point between t+a and t+b.
If Boxplus[a,b]A is true at the time t, it requires that A needs to be true continuously between t+a and t+b.
You should not give any explanation and you should only output "true" or "false" | You are given a dataset and a temporal rule, and your task is to judge whether the given fact is entailed by the dataset and the rule.
The rules are expressed as DatalogMTL, a knowledge representation language that extends Datalog with operators from metric temporal logic (MTL). The semantics of four MTL operators are given as follows:
If Diamondminus[a,b]A is true at the time t, it requires that A needs to be true at some time between t-b and t-a.
If Boxminus[a,b]A is true at the time t, it requires that A needs to be true continuously between t-b and t-a.
If Diamondplus[a,b]A is true at the time t, it requires that A needs to be true at some point between t+a and t+b.
If Boxplus[a,b]A is true at the time t, it requires that A needs to be true continuously between t+a and t+b. |
| User Prompt | Now we have some temporal data and some rules, data:
B@[6,9]
rule: A:-Diamondminus[6,15]B
Is A@[12,18] true or not? | Now we have some temporal data and some rules, data:
B@[6,9]
rule: A:-Diamondminus[6,15]B
Is A@[12,18] true or not? Do not answer directly, think step by step. |

| LLM's output | | First, let's understand the rule: A:-Diamondminus[6,15]B. This rule means that if B is true at some point between t-15 and t-6, then A is true at time t. |
|---|---|---|
| | | Now, let's consider the fact A@[12,18]. This means we are checking if A is true at some point between time 12 and 18. |
| | | To check this, we need to see if B is true at some point between (12-15) and (12-6), i.e., between time -3 and 6, or between (18-15) and (18-6), i.e., between time 3 and 12. |
| | | From the given data, we know that B is true between time 6 and 9. This falls within the required time range for A to be true. |
| | | Therefore, A@[12,18] is true. |
| User Prompt | | Based on your previous response, without any explanation state your answers to the question. You should only output "true" or "false" |
| LLM's output | false | true |
| Expected Answer | true | |

Example 2:

| | Without CoT | With CoT |
|---|---|---|
| System Prompt | You are given a dataset and a temporal rule, and your task is to judge whether the given fact is entailed by the dataset and the rule. | You are given a dataset and a temporal rule, and your task is to judge whether the given fact is entailed by the dataset and the rule. |
| | The rules are expressed as DatalogMTL, a knowledge representation language that extends Datalog with operators from metric temporal logic (MTL). The semantics of four MTL operators are given as follows: | The rules are expressed as DatalogMTL, a knowledge representation language that extends Datalog with operators from metric temporal logic (MTL). The semantics of four MTL operators are given as follows: |
| | If Diamondminus[a,b]A is true at the time t, it requires that A needs to be true at some time between t-b and t-a. | If Diamondminus[a,b]A is true at the time t, it requires that A needs to be true at some time between t-b and t-a. |
| | If Boxminus[a,b]A is true at the time t, it requires that A needs to be true continuously between t-b and t-a. | If Boxminus[a,b]A is true at the time t, it requires that A needs to be true continuously between t-b and t-a. |
| | If Diamondplus[a,b]A is true at the time t, it requires that A needs to be true at some point between t+a and t+b. | If Diamondplus[a,b]A is true at the time t, it requires that A needs to be true at some point between t+a and t+b. |
| | If Boxplus[a,b]A is true at the time t, it requires that A needs to be true continuously between t+a and t+b. | If Boxplus[a,b]A is true at the time t, it requires that A needs to be true continuously between t+a and t+b. |
| | You should not give any explanation and you should only output "true" or "false" | |

| User Prompt | Now we have some temporal data and some rules, data:
A@[8,13]
rule: B:-Diamondminus[10,12]A
Is B@[21,24] true or not? | Now we have some temporal data and some rules, data:
A@[8,13]
rule: B:-Diamondminus[10,12]A
Is B@[21,24] true or not? Do not answer directly, think step by step. |
|---|---|---|
| LLM's output | | First, let's understand the rule: B:-Diamondminus[10,12]A. This rule means that if A is true at some point between t-12 and t-10, then B is true at time t.
Now, let's look at the data: A@[8,13]. This means that A is true continuously from time 8 to time 13.
Now, we need to check if B@[21,24] is true or not. For B to be true at any time between 21 and 24, A needs to be true at some point between (t-12) and (t-10).
Let's check for each time t in [21,24]:
- For t=21, A needs to be true at some point between 21-12=9 and 21-10=11. A is true at these times according to the data.
- For t=22, A needs to be true at some point between 22-12=10 and 22-10=12. A is true at these times according to the data.
- For t=23, A needs to be true at some point between 23-12=11 and 23-10=13. A is true at these times according to the data.
- For t=24, A needs to be true at some point between 24-12=12 and 24-10=14. A is true at time 12, but not at time 14. However, the rule requires A to be true at some point between these times, not at all times, so this condition is satisfied.
Therefore, B@[21,24] is true. |
| User Prompt | | Based on your previous response, without any explanation state your answers to the question. You should only output "true" or "false" |
| LLM's output | false | true |
| Expected Answer | true | |

## D.3 SINGLEATOM ERROR CASES

False Negative Example:

| System Prompt | You are given a dataset and a temporal rule, and your task is to judge whether the given fact is entailed by the dataset and the rule. The rules are expressed as DatalogMTL, a knowledge representation language that extends Datalog with operators from metric temporal logic (MTL). The semantics of four MTL operators are given as follows:If Diamondminus[a,b]A is true at the time t, it requires that A needs to be true at some time between t-b and t-a.If Boxminus[a,b]A is true at the time t, it requires that A needs to be true continuously between t-b and t-a.If Diamondplus[a,b]A is true at the time t, it requires that A needs to be true at some point between t+a and t+b.If Boxplus[a,b]A is true at the time t, it requires that A needs to be true continuously between t+a and t+b. |
|---|---|
| User Prompt | Now we have some temporal data and some rules, data: A@[7,15] 
 rule: B:-Boxminus[3,5]A 
 Is B@[8,30] true or not? Do not answer directly, think step by step. |
| LLM's output | To determine whether ( B@[8,30] ) is true given the data and the rule, we need to follow these steps: 
 1. **Understand the Rule**: The rule is ( B leftarrow Boxminus[3,5]A ). This means that ( B ) is true at time ( t ) if ( A ) is true continuously in the interval ([t-5, t-3]). 
 2. **Check the Interval for ( B@[8,30] )**: We need to check if ( B ) is true for every time point in the interval ([8, 30]). For each time ( t ) in this interval, ( A ) must be true continuously in the interval ([t-5, t-3]). 
 3. **Analyze the Data**: The data given is ( A@[7,15] ), which means ( A ) is true continuously from time 7 to time 15. 
 4. **Check Each Time Point in ([8, 30])**: 
 - For ( t = 8 ): The interval ([t-5, t-3]) is ([3, 5]). ( A ) is not true in this interval because ( A ) is only true from 7 to 15. 
 - For ( t = 9 ): The interval ([t-5, t-3]) is ([4, 6]). ( A ) is not true in this interval. 
 - For ( t = 10 ): The interval ([t-5, t-3]) is ([5, 7]). ( A ) is not true continuously in this interval (it starts at 7). 
 - For ( t = 11 ): The interval ([t-5, t-3]) is ([6, 8]). ( A ) is true continuously in this interval. 
 - For ( t = 12 ): The interval ([t-5, t-3]) is ([7, 9]). ( A ) is true continuously in this interval. 
 - For ( t = 13 ): The interval ([t-5, t-3]) is ([8, 10]). ( A ) is true continuously in this interval. 
 - For ( t = 14 ): The interval ([t-5, t-3]) is ([9, 11]). ( A ) is true continuously in this interval. 
 - For ( t = 15 ): The interval ([t-5, t-3]) is ([10, 12]). ( A ) is true continuously in this interval. 
 - For ( t = 16 ): The interval ([t-5, t-3]) is ([11, 13]). ( A ) is true continuously in this interval. 
 - For ( t = 17 ): The interval ([t-5, t-3]) is ([12, 14]). ( A ) is true continuously in this interval. 
 - For ( t = 18 ): The interval ([t-5, t-3]) is ([13, 15]). ( A ) is true continuously in this interval. 
 - For ( t = 19 ): The interval ([t-5, t-3]) is ([14, 16]). ( A ) is not true in this interval (it ends at 15). 
 - For ( t = 20 ) to ( t = 30 ): The intervals ([t-5, t-3]) will be ([15, 17]) to ([25, 27]). ( A ) is not true in any of these intervals. 
 5. **Conclusion**: ( B ) is not true for all time points in the interval ([8, 30]). Specifically, ( B ) is not true at ( t = 8, 9, 10, 19, ) and any ( t ) from 20 to 30. Therefore, ( B@[8,30] ) is not true. |
| User Prompt | Based on your previous response, without any explanation state your answers to the question. You should only output "true" or "false" |
| LLM's output | false |
| Expected Answer | true |

False Positive Example:

| System Prompt | You are given a dataset and a temporal rule, and your task is to judge whether the given fact is entailed by the dataset and the rule. The rules are expressed as DatalogMTL, a knowledge representation language that extends Datalog with operators from metric temporal logic (MTL). The semantics of four MTL operators are given as follows:If Diamondminus[a,b]A is true at the time t, it requires that A needs to be true at some time between t-b and t-a.If Boxminus[a,b]A is true at the time t, it requires that A needs to be true continuously between t-b and t-a.If Diamondplus[a,b]A is true at the time t, it requires that A needs to be true at some point between t+a and t+b.If Boxplus[a,b]A is true at the time t, it requires that A needs to be true continuously between t+a and t+b. |
|---|---|
| User Prompt | Now we have some temporal data and some rules, data: A@[0,3] rule: B:-Diamondminus[2,4]A Is B@[-34,8] true or not? Do not answer directly, think step by step. |
| LLM's output | To determine whether ( B@[-34,8] ) is true given the data and the rule, we need to follow these steps: 1. **Understand the Rule**: The rule is ( B leftarrow Diamondminus[2,4]A ). This means that ( B ) is true at time ( t ) if ( A ) is true at some time between ( t-4 ) and ( t-2 ). 2. **Determine the Time Interval for ( B@[-34,8] )**: We need to check if ( B ) is true at any time within the interval ([-34, 8]). 3. **Apply the Rule to Each Time Point in the Interval**: - For each time ( t ) in the interval ([-34, 8]), we need to check if ( A ) is true at some time between ( t-4 ) and ( t-2 ). 4. **Check the Data**: The data given is ( A@[0,3] ), which means ( A ) is true from time 0 to time 3. 5. **Evaluate the Rule for Each Time Point**: - For ( t = -34 ): We need ( A ) to be true between ([-38, -36]). Since ( A ) is only true between ([0, 3]), ( A ) is not true in this interval. - For ( t = -33 ): We need ( A ) to be true between ([-37, -35]). Again, ( A ) is not true in this interval. - Continue this process up to ( t = 8 ): - For ( t = 0 ): We need ( A ) to be true between ([-4, -2]). ( A ) is not true in this interval. - For ( t = 1 ): We need ( A ) to be true between ([-3, -1]). ( A ) is not true in this interval. - For ( t = 2 ): We need ( A ) to be true between ([-2, 0]). ( A ) is not true in this interval. - For ( t = 3 ): We need ( A ) to be true between ([-1, 1]). ( A ) is true at time 0, which is within this interval. - For ( t = 4 ): We need ( A ) to be true between ([0, 2]). ( A ) is true in this interval. - For ( t = 5 ): We need ( A ) to be true between ([1, 3]). ( A ) is true in this interval. - For ( t = 6 ): We need ( A ) to be true between ([2, 4]). ( A ) is true at time 2 and 3, which is within this interval. - For ( t = 7 ): We need ( A ) to be true between ([3, 5]). ( A ) is true at time 3, which is within this interval. - For ( t = 8 ): We need ( A ) to be true between ([4, 6]). ( A ) is not true in this interval. 6. **Conclusion**: Since ( A ) is true at some time between ( t-4 ) and ( t-2 ) for ( t = 3, 4, 5, 6, ) and ( 7 ), ( B ) is true at these times. Therefore, ( B ) is true at some points within the interval ([-34, 8]). Thus, ( B@[-34,8] ) is true. |
| User Prompt | Based on your previous response, without any explanation state your answers to the question. You should only output "true" or "false" |
| LLM's output | true |
| Expected Answer | false |

## D.4 COMPARISON BETWEEN COT PROMPT AND DEEPSEEK REASONING

Example 1:

Table 9: A detailed context comparison between GPT-4o with CoT and DeepSeek-R1-Distill-Qwen-32B

| | Qwen2.5-32B-Instruct with CoT | DeepSeek-R1-Distill-Qwen-32B |
|---|---|---|
| System Prompt | You are given a dataset and a temporal rule, and your task is to judge whether the given fact is entailed by the dataset and the rule. The rules are expressed as Data-logMTL, a knowledge representation language that extends Datalog with operators from metric temporal logic (MTL). The semantics of four MTL operators are given as follows: If Diamondminus[a,b]A is true at the time t, it requires that A needs to be true at some time between t-b and t-a. If Boxminus[a,b]A is true at the time t, it requires that A needs to be true continuously between t-b and t-a. If Diamondplus[a,b]A is true at the time t, it requires that A needs to be true at some point between t+a and t+b. If Boxplus[a,b]A is true at the time t, it requires that A needs to be true continuously between t+a and t+b. | You are given a dataset and a temporal rule, and your task is to judge whether the given fact is entailed by the dataset and the rule. The rules are expressed as Data-logMTL, a knowledge representation language that extends Datalog with operators from metric temporal logic (MTL). The semantics of four MTL operators are given as follows: If Diamondminus[a,b]A is true at the time t, it requires that A needs to be true at some time between t-b and t-a. If Boxminus[a,b]A is true at the time t, it requires that A needs to be true continuously between t-b and t-a. If Diamondplus[a,b]A is true at the time t, it requires that A needs to be true at some point between t+a and t+b. If Boxplus[a,b]A is true at the time t, it requires that A needs to be true continuously between t+a and t+b. You should not give any explanation and you should only output "true" or "false" |
| User Prompt | Now we have some temporal data and some rules, data: A@[9,12] rule: B:-Diamondminus[3,13]A Is B@[25,25] true or not? Do not answer directly, think step by step. | Now we have some temporal data and some rules, data: A@[9,12] rule: B:-Diamondminus[3,13]A Is B@[25,25] true or not? |

| | | |
|---|---|---|
| LLM's output | To determine if B@[25,25] is true given the rule and the data, we need to follow these steps:
1. **Understand the Rule**: The rule is B:-Diamondminus[3,13]A. This means that B is true at a time t if A is true at some point between t-13 and t-3.
2. **Identify the Time Interval for B**: We are checking if B@[25,25] is true. This means we are looking at the time t = 25.
3. **Determine the Relevant Time Interval for A**: According to the rule, for B to be true at t = 25, A must be true at some point between 25-13 and 25-3, which is between 12 and 22.
4. **Check the Data for A**: The data provided is A@[9,12]. This means A is true from time 9 to time 12.
5. **Compare the Intervals**: The interval for A (9 to 12) does not overlap with the interval required by the rule for B (12 to 22). Specifically, A is only true up to time 12, and the rule requires A to be true at some point between 12 and 22.
6. **Conclusion**: Since A is not true at any point between 12 and 22, B@[25,25] cannot be true based on the given rule and data.
Therefore, B@[25,25] is not true. | Okay, so I need to figure out if B is true at time 25 based on the given data and rule. Let me start by understanding the problem step by step.
First, the data provided is A@[9,12]. This means that A is true from time 9 up to and including time 12. So, A is active during the interval [9,12].
Next, the rule given is B:-Diamondminus[3,13]A. I need to parse this rule correctly. In DatalogMTL, the Diamondminus operator is used, which has a specific meaning. From the semantics provided, Diamondminus[a,b]A being true at time t means that A must be true at some time between t - b and t - a.
So, breaking down the rule: B is derived if there exists some time within the interval [t - 13, t - 3] where A is true. Therefore, for B to hold at time t, A must be true at least once in that interval.
Now, the question is whether B is true at time 25, specifically at the point [25,25]. So, t is 25. Let's compute the interval for A based on the rule.
The interval for A would be from t - b to t - a, which is 25 - 13 to 25 - 3. Calculating that:
25 - 13 = 12 25 - 3 = 22
So, the interval is [12,22]. This means that for B to be true at 25, A must be true at some point between 12 and 22.
Looking back at the data, A is true from 9 to 12. So, A is true at 12, which is the end of its interval. Now, does 12 fall within [12,22]? Yes, it does. Therefore, A is true at 12, which is within the required interval for B to hold at 25.
Wait, but I should double-check. The interval for A is [9,12], so A is true at 12. The rule requires A to be true at some point between 12 and 22. Since 12 is included in both intervals, A is true at 12, which satisfies the condition.
Therefore, B should be true at time 25. |
| User Prompt | Based on your previous response, without any explanation state your answers to the question. You should only output "true" or "false" | |
| LLM's output | false | true |

| Expected Answer | true |
|---|---|

Example 2:

Table 10: A detailed context comparison between GPT-4o with CoT and DeepSeek-R1-Distill-Qwen-32B

| | Qwen2.5-32B-Instruct with CoT | DeepSeek-R1-Distill-Qwen-32B |
|---|---|---|
| System Prompt | You are given a dataset and a temporal rule, and your task is to judge whether the given fact is entailed by the dataset and the rule.
The rules are expressed as DatalogMTL, a knowledge representation language that extends Datalog with operators from metric temporal logic (MTL). The semantics of four MTL operators are given as follows:
If Diamondminus[a,b]A is true at the time t, it requires that A needs to be true at some time between t-b and t-a.
If Boxminus[a,b]A is true at the time t, it requires that A needs to be true continuously between t-b and t-a.
If Diamondplus[a,b]A is true at the time t, it requires that A needs to be true at some point between t+a and t+b.
If Boxplus[a,b]A is true at the time t, it requires that A needs to be true continuously between t+a and t+b. | You are given a dataset and a temporal rule, and your task is to judge whether the given fact is entailed by the dataset and the rule.
The rules are expressed as DatalogMTL, a knowledge representation language that extends Datalog with operators from metric temporal logic (MTL). The semantics of four MTL operators are given as follows:
If Diamondminus[a,b]A is true at the time t, it requires that A needs to be true at some time between t-b and t-a.
If Boxminus[a,b]A is true at the time t, it requires that A needs to be true continuously between t-b and t-a.
If Diamondplus[a,b]A is true at the time t, it requires that A needs to be true at some point between t+a and t+b.
If Boxplus[a,b]A is true at the time t, it requires that A needs to be true continuously between t+a and t+b.
You should not give any explanation and you should only output "true" or "false" |
| User Prompt | Now we have some temporal data and some rules, data:
A@[6,6]
rule: A:-Diamondminus[2,2]A
Is A@[80,80] true or not? Do not answer directly, think step by step. | Now we have some temporal data and some rules, data:
A@[6,6]
rule: A:-Diamondminus[2,2]A
Is A@[80,80] true or not? |

| LLM's output | To determine whether A@[80,80] is true given the rule A:-Diamondminus[2,2]A and the data A@[6,6], we need to follow these steps: | Okay, so I'm trying to figure out whether A@[80,80] is true based on the given data and rule. Let me break this down step by step. |
|---|---|---|
| | 1. **Understand the Rule**: The rule A:-Diamondminus[2,2]A means that A is true at time t if A is true at some point between t-2 and t+2. | First, the data provided is A@[6,6]. That means at time 6, A is true. Now, the rule is A:-Diamondminus[2,2]A. I need to understand what this rule means. |
| | 2. **Analyze the Data**: The data provided is A@[6,6], which means A is true at time 6. | The Diamondminus operator in MTL is defined such that Diamondminus[a,b]A is true at time t if A is true at some time between t-b and t-a. So, for the rule A:-Diamondminus[2,2]A, it means that A is true at time t if there exists some time between t-2 and t-2 (since a and b are both 2) where A is true. |
| | 3. **Apply the Rule to the Data**: According to the rule, A@[t] is true if A is true at some point between t-2 and t+2. We need to check if A@[6,6] can satisfy the rule for any time t. | Wait, that seems a bit confusing. Let me clarify. The operator Diamondminus[2,2]A at time t requires that A is true at some point between t-2 and t-2. But t-2 and t-2 is just a single point in time, t-2. So, effectively, this rule is saying that A is true at time t if A was true at time t-2. |
| | 4. **Check the Time Interval for A@[80,80]**: We need to see if A@[80,80] can be true based on the rule. For A@[80,80] to be true, A must be true at some point between 80-2 and 80+2, i.e., between 78 and 82. | So, applying this rule, if A is true at time t-2, then A is true at time t. Now, looking at the data, A is true at time 6. So, according to the rule, A should also be true at time 6 + 2 = 8. Then, applying the rule again, A at 8 would imply A at 10, and so on. This creates a chain where A is true at every even time starting from 6: 6, 8, 10, 12, etc. |
| | 5. **Compare the Given Data with the Required Interval**: The data A@[6,6] indicates that A is true at time 6. However, this does not fall within the interval [78, 82]. | But the question is about A@[80,80]. Is 80 an even number? Yes, 80 is even. So, following the chain, A should be true at 80 because it's 6 + 2*37 = 80. Therefore, A is true at 80. |
| | 6. **Conclusion**: Since A@[6,6] does not fall within the interval [78, 82], A@[80,80] cannot be true based on the given rule and data. Therefore, A@[80,80] is not true. | Wait, but let me double-check. The rule is A:-Diamondminus[2,2]A, which as I understood, means A at t if A was at t-2. So starting from 6, A is true at 8, then 10, and so on. So 80 is 6 + 2*37, which is correct. So yes, A should be true at 80. |
| | | I think that's the correct reasoning. So the answer should be true. |
| User Prompt | Based on your previous response, without any explanation state your answers to the question. You should only output "true" or "false" | |
| LLM's output | false | true |
| Expected Answer | true | |

### D.5 A SYSTEMATIC TAXONOMY OF LLM FAILURES

We devised a method that uses the powerful Gemini-2.5-Pro model as a proxy to analyze the chain-of-thought (CoT) processes in failure cases generated by Qwen 2.5 32B. This automated analysis revealed six recurring categories of errors:

**1. Failure to apply rules recursively**

| | |
|---|---|
| System Prompt | You are given a dataset and a temporal rule, and your task is to judge whether the given fact is entailed by the dataset and the rule. The rules are expressed as DatalogMTL, a knowledge representation language that extends Datalog with operators from metric temporal logic (MTL). The semantics of four MTL operators are given as follows:If Diamondminus[a,b]A is true at the time t, it requires that A needs to be true at some time between t-b and t-a.If Boxminus[a,b]A is true at the time t, it requires that A needs to be true continuously between t-b and t-a.If Diamondplus[a,b]A is true at the time t, it requires that A needs to be true at some point between t+a and t+b.If Boxplus[a,b]A is true at the time t, it requires that A needs to be true continuously between t+a and t+b. |
| User Prompt | Now we have some temporal data and some rules, data: A@[4,14]
rule: A:-Diamondminus[1,8]A
Is A@[218,365] true or not? Do not answer directly, think step by step. |
| LLM's output | To determine if A@[218,365] is true given the rule A:-Diamondminus[1,8]A and the data A@[4,14], we need to follow these steps:
1. **Understand the Rule**: The rule A:-Diamondminus[1,8]A means that A is true at a time t if A is true at some point between t-8 and t-1.
2. **Analyze the Data**: The data provided is A@[4,14], which means A is true between time 4 and time 14.
3. **Apply the Rule to the Given Time Interval**: We need to check if A@[218,365] can be true based on the rule and the data.
- For A@[218,365] to be true, there must be some time t between 218 and 365 where A is true at some point between t-8 and t-1. - We need to check if the interval [218-8, 218-1] = [210, 217] or any other interval within [218, 365] intersects with the interval [4, 14] where A is true.
4. **Check for Intersection**: - The interval [210, 217] does not intersect with [4, 14]. - Similarly, any interval [t-8, t-1] for t in [218, 365] will not intersect with [4, 14].
Since there is no overlap between the interval [4, 14] and any interval [t-8, t-1] for t in [218, 365], the rule A:-Diamondminus[1,8]A cannot be satisfied for A@[218,365].
Therefore, A@[218,365] is not true based on the given rule and data. |
| User Prompt | Based on your previous response, without any explanation state your answers to the question. You should only output "true" or "false" |
| LLM's output | false |
| Expected Answer | true |

The reasoning process fails to account for the recursive application of the rule. It performs only a single-step check and does not apply the rule to its own conclusions, thus missing the chain of derivations that would prove the query.

**2. Incorrect interval overlap or boundary check**

| System Prompt | You are given a dataset and a temporal rule, and your task is to judge whether the given fact is entailed by the dataset and the rule. The rules are expressed as DatalogMTL, a knowledge representation language that extends Datalog with operators from metric temporal logic (MTL). The semantics of four MTL operators are given as follows:If Diamondminus[a,b]A is true at the time t, it requires that A needs to be true at some time between t-b and t-a.If Boxminus[a,b]A is true at the time t, it requires that A needs to be true continuously between t-b and t-a.If Diamondplus[a,b]A is true at the time t, it requires that A needs to be true at some point between t+a and t+b.If Boxplus[a,b]A is true at the time t, it requires that A needs to be true continuously between t+a and t+b. |
|---|---|
| User Prompt | Now we have some temporal data and some rules, data: A@[9,12]
rule: B:-Diamondminus[3,13]A
Is B@[25,25] true or not? Do not answer directly, think step by step. |
| LLM's output | To determine if B@[25,25] is true given the rule and the data, we need to follow these steps:
1. **Understand the Rule**: The rule is B:-Diamondminus[3,13]A. This means that B is true at a time t if A is true at some point between t-13 and t-3.
2. **Identify the Time Interval for B**: We are checking if B@[25,25] is true. This means we are looking at the time t = 25.
3. **Determine the Relevant Time Interval for A**: According to the rule, for B to be true at t = 25, A must be true at some point between 25-13 and 25-3, which is between 12 and 22.
4. **Check the Data for A**: The data provided is A@[9,12]. This means A is true from time 9 to time 12.
5. **Compare the Intervals**: The interval for A (9 to 12) does not overlap with the interval required by the rule for B (12 to 22). Specifically, A is only true up to time 12, and the rule requires A to be true at some point between 12 and 22.
6. **Conclusion**: Since A is not true at any point between 12 and 22, B@[25,25] cannot be true based on the given rule and data.
Therefore, B@[25,25] is not true. |
| User Prompt | Based on your previous response, without any explanation state your answers to the question. You should only output "true" or "false" |
| LLM's output | false |
| Expected Answer | true |

The reasoning claims there is no overlap between the interval where $A$ is true ($[9, 12]$) and the interval required by the rule ($[12, 22]$). This is incorrect. The two intervals intersect at the single time point $t = 12$.

### 3. Misinterpretation of query semantics

| System Prompt | You are given a dataset and a temporal rule, and your task is to judge whether the given fact is entailed by the dataset and the rule. The rules are expressed as DatalogMTL, a knowledge representation language that extends Datalog with operators from metric temporal logic (MTL). The semantics of four MTL operators are given as follows:If Diamondminus[a,b]A is true at the time t, it requires that A needs to be true at some time between t-b and t-a.If Boxminus[a,b]A is true at the time t, it requires that A needs to be true continuously between t-b and t-a.If Diamondplus[a,b]A is true at the time t, it requires that A needs to be true at some point between t+a and t+b.If Boxplus[a,b]A is true at the time t, it requires that A needs to be true continuously between t+a and t+b. |
|---|---|

| | |
|---|---|
| User Prompt | Now we have some temporal data and some rules, data: A@[0,15] B@[6,10] rule: C:-Diamondminus[2,15]A,Boxminus[6,9]B Is C@[-4,19] true or not? Do not answer directly, think step by step. |
| LLM's output | To determine whether C@[-4,19] is true given the data and the rule, we need to break down the rule and check if the conditions are met for each time point in the interval [-4,19]. The rule is: C :- Diamondminus[2,15]A, Boxminus[6,9]B This means that C is true at a time t if: 1. There exists some time between t-15 and t-2 where A is true (Diamondminus[2,15]A). 2. B is true continuously between t-9 and t-6 (Boxminus[6,9]B). Let's analyze the data: - A is true in the interval [0,15]. - B is true in the interval [6,10]. Now, let's check the conditions for each time point in the interval [-4,19]. ### Condition 1: Diamondminus[2,15]A This condition requires that A is true at some point between t-15 and t-2. We need to check if this condition holds for each t in [-4,19]. - For t = -4: The interval is [-19, -6]. A is not true in this interval. - For t = 19: The interval is [4, 17]. A is true in this interval. Since A is true in the interval [0,15], we need to find the range of t where the interval [t-15, t-2] overlaps with [0,15]. - The lower bound of the interval [t-15, t-2] should be less than or equal to 15: t-15 <= 15 => t <= 30. - The upper bound of the interval [t-15, t-2] should be greater than or equal to 0: t-2 >= 0 => t >= 2. So, the condition Diamondminus[2,15]A is true for t in the interval [2, 30]. ### Condition 2: Boxminus[6,9]B This condition requires that B is true continuously between t-9 and t-6. We need to check if this condition holds for each t in [-4,19]. - For t = -4: The interval is [-13, -10]. B is not true in this interval. - For t = 19: The interval is [10, 13]. B is not true in this interval. Since B is true in the interval [6,10], we need to find the range of t where the interval [t-9, t-6] is within [6,10]. - The lower bound of the interval [t-9, t-6] should be greater than or equal to 6: t-9 >= 6 => t >= 15. - The upper bound of the interval [t-9, t-6] should be less than or equal to 10: t-6 <= 10 => t <= 16. So, the condition Boxminus[6,9]B is true for t in the interval [15, 16]. ### Combining the Conditions To satisfy the rule, both conditions must be true simultaneously. Therefore, we need to find the intersection of the intervals [2, 30] and [15, 16]. The intersection of [2, 30] and [15, 16] is [15, 16]. ### Conclusion C is true for t in the interval [15, 16]. Since the interval [15, 16] is a subset of [-4, 19], C@[-4,19] is true for the time points in [15, 16]. Therefore, C@[-4,19] is true. |
| User Prompt | Based on your previous response, without any explanation state your answers to the question. You should only output "true" or "false" |
| LLM's output | true |
| Expected Answer | false |

This is incorrect. For $C@[-4, 19]$ to be true, $C$ must be true for all time points from -4 to 19. Since $C$ is only true for the interval $[15, 16]$, it is false for other parts of the query interval, such as $[-4, 15)$ and $(16, 19]$. Therefore, the query $C@[-4, 19]$ is false. The reasoning mistakenly assumes that if $C$ is true for any part of the query interval, the query itself is true.

**4. Incorrect interval calculation**

| System Prompt | You are given a dataset and a temporal rule, and your task is to judge whether the given fact is entailed by the dataset and the rule. The rules are expressed as DatalogMTL, a knowledge representation language that extends Datalog with operators from metric temporal logic (MTL). The semantics of four MTL operators are given as follows:If Diamondminus[a,b]A is true at the time t, it requires that A needs to be true at some time between t-b and t-a.If Boxminus[a,b]A is true at the time t, it requires that A needs to be true continuously between t-b and t-a.If Diamondplus[a,b]A is true at the time t, it requires that A needs to be true at some point between t+a and t+b.If Boxplus[a,b]A is true at the time t, it requires that A needs to be true continuously between t+a and t+b. |
|---|---|
| User Prompt | Now we have some temporal data and some rules, data: B@[2.795,14.633] rule: A:-Boxplus[5.046,10.683]B 
 Is A@[-0.210,1.972] true or not? Do not answer directly, think step by step. |
| LLM's output | To determine if A@[-0.210,1.972] is true given the rule A:-Boxplus[5.046,10.683]B and the data B@[2.795,14.633], we need to follow these steps: 
 1. **Understand the Rule**: The rule A:-Boxplus[5.046,10.683]B means that A is true at a time t if B is true continuously between t+5.046 and t+10.683. 
 2. **Analyze the Data**: The data B@[2.795,14.633] means that B is true between time 2.795 and 14.633. 
 3. **Determine the Time Interval for A**: We need to find the time interval for A such that B is true continuously between t+5.046 and t+10.683. This means we need to find the time t such that the interval [t+5.046, t+10.683] is within the interval [2.795, 14.633]. 
 4. **Calculate the Interval for t**: - The lower bound of the interval for B is 2.795, so t+5.046 <= 2.795. - The upper bound of the interval for B is 14.633, so t+10.683 <= 14.633. 
 Solving these inequalities: - t+5.046 <= 2.795 => t <= 2.795 - 5.046 => t <= -2.251 - t+10.683 <= 14.633 => t <= 14.633 - 10.683 => t <= 3.950 
 Since t must satisfy both conditions, we take the stricter condition: - t <= -2.251 
 5. **Check the Given Interval for A**: We need to check if the interval [-0.210, 1.972] satisfies the condition t <= -2.251. Clearly, the interval [-0.210, 1.972] does not satisfy this condition because -0.210 is greater than -2.251. 
 Therefore, A@[-0.210,1.972] is not true based on the given rule and data. |
| User Prompt | Based on your previous response, without any explanation state your answers to the question. You should only output "true" or "false" |
| LLM's output | false |
| Expected Answer | true |

LLM says that $t + 5.046 \leq 2.795$, which is incorrect. The corret one should be $t + 5.046 \geq 2.795$.

**5. Misinterpretation of operator semantics**

| System Prompt | You are given a dataset and a temporal rule, and your task is to judge whether the given fact is entailed by the dataset and the rule. The rules are expressed as DatalogMTL, a knowledge representation language that extends Datalog with operators from metric temporal logic (MTL). The semantics of four MTL operators are given as follows:If Diamondminus[a,b]A is true at the time t, it requires that A needs to be true at some time between t-b and t-a.If Boxminus[a,b]A is true at the time t, it requires that A needs to be true continuously between t-b and t-a.If Diamondplus[a,b]A is true at the time t, it requires that A needs to be true at some point between t+a and t+b.If Boxplus[a,b]A is true at the time t, it requires that A needs to be true continuously between t+a and t+b. |
|---|---|
| User Prompt | Now we have some temporal data and some rules, data: B@[2,12] rule: C:-Diamondplus[5,8]A A:-Boxplus[10,12]B Is C@[-5,-5] true or not? Do not answer directly, think step by step. |
| LLM's output | To determine whether C@[-5,-5] is true given the data and the rules, we need to follow the rules step by step and see if the conditions are met. 1. **Data Provided:** - B@[2,12]: This means B is true continuously from time 2 to time 12. 2. **Rules Provided:** - C :- Diamondplus[5,8]A: This means C is true at time t if A is true at some point between t+5 and t+8. - A :- Boxplus[10,12]B: This means A is true at time t if B is true continuously between t+10 and t+12. 3. **Determine A's Truth Value:** - We need to check if A is true at some time t such that B is true continuously between t+10 and t+12. - Given B@[2,12], B is true from time 2 to time 12. - For A to be true at time t, B must be true continuously from t+10 to t+12. - The latest time B is true is 12, so t+12 must be less than or equal to 12. - Therefore, t must be less than or equal to 0 (t+12 <= 12 => t <= 0). - The earliest time B is true is 2, so t+10 must be greater than or equal to 2. - Therefore, t must be greater than or equal to -8 (t+10 >= 2 => t >= -8). - Combining these, A is true for t in the range [-8, 0]. 4. **Determine C's Truth Value:** - We need to check if C is true at time t such that A is true at some point between t+5 and t+8. - A is true for t in the range [-8, 0]. - For C to be true at time t, A must be true at some point between t+5 and t+8. - We need to find the range of t such that t+5 to t+8 overlaps with [-8, 0]. - The latest time A is true is 0, so t+8 must be less than or equal to 0. - Therefore, t must be less than or equal to -8 (t+8 <= 0 => t <= -8). - The earliest time A is true is -8, so t+5 must be greater than or equal to -8. - Therefore, t must be greater than or equal to -13 (t+5 >= -8 => t >= -13). - Combining these, C is true for t in the range [-13, -8]. 5. **Check C@[-5,-5]:** - We need to check if C is true at time -5. - From the previous step, C is true for t in the range [-13, -8]. - Since -5 is not in the range [-13, -8], C is not true at time -5. Therefore, C@[-5,-5] is not true. |
| User Prompt | Based on your previous response, without any explanation state your answers to the question. You should only output "true" or "false" |
| LLM's output | false |
| Expected Answer | true |

The reasoning incorrectly applies the logic of a Box operator instead of a Diamond operator.

## E CAN LLM UNDERSTAND SYMBOLIC REPRESENTATION?

We ask LLM to translate generated natural language representations of logic rules to symbolic representations and them compare the translated symbolic representations with the ground truth symbolic representation to verify if LLM has the ability to understand symbolic representations. Specifically, we passed the same prompt that used in our symbolic evaluations, "The rules are expressed as Data-

logMTL, a......If Boxplus[a,b]A is true at the time t, it requires that A needs to be true continuously between t+a and t+b.", into LLMs, along with few examples telling LLM the output format, then ask LLM to translate verbalized samples into symbolic ones. If the translated symbolic rule from the verbalized rule is exactly the same as the original symbolic rule, then we consider LLM has the ability to understand both the symbolic rule and the verbalized rule.

We passed 6x50 samples selected from all six subsets into the LLMs. LLM accurately translated 100% of testing samples for most subsets, and 96% of testing samples for MultiRule subset, from verbalized representations to symbolic representations.

In addition, we noticed that larger LLMs with strong reasoning abilities, such as DeepSeek-R1, performs pertty good on some cases, further proving that the semantics is understood.

Considering all those points, We believe that LLM can understand the symbolic representation.

## F    DETAILED BENCHMARK CONSTRUCTION PSEUDO CODE

Our dataset generation algorithm is driven by generating rules. In a high level view, it generate rules one by one in a same context, while the generation process for each rule contains the context check, ensuring the generated rules are non-trivial.

---

**Algorithm 1:** Generate

---

**Parameters:** $f$: The set of features Enabled
**Parameters:** $N$: The number of rules
**Parameters:** $V$: A boolean flag to control if the program should generate a positive sample or a
                 negative sample
**Output:** A problem instance $I$ containing a set of rules, a set of data, a query and a boolean
         value representing whether the query is valid or not.
$G \leftarrow EmptyGraph()$;
**while** $i$ in 1.....N **do**
    **do**
        $G \leftarrow GenerateGraph(G)$;
        **while** $n$ in $G.nodes$ **do**
            Assign node with random values
        **end**
        $G \leftarrow GenerateRules(G)$
    **while** *New Info can be Inferred from I*;
**end**
$Rules, Data \leftarrow Extract\ Rules\ associed\ with\ G$;
$DeltaNew \leftarrow Facts\ Inferred\ From\ Graph\ G$;
$QueryEntity, Interval \leftarrow$ Randomly Select From $DeltaNew$;
**if** *V* **then**
    $QueryInterval \leftarrow$ A random sub-interval from $Interval$;
**else**
    $QueryInterval \leftarrow$ A random sub-interval that is not in $Interval$;
**end**
**return** *Rules, Data, QueryEntity, QueryInterval, V*

---

The graph generation algorithm 2 will generate a graph where nodes in the graph represents predicates such as A, B and C. We are going to attach details information about predicates and rules into the corresponding nodes and edges of the graph, but at this time we only need the structure of the graph, i.e. nodes and edges don't have special information attached.

---

**Algorithm 2:** Graph Generation

---

**Input:** $G$: The existing graph
**Parameters:** $f$: The set of features Enabled
**Output:** $G$: The generated graph (including the old information in the existing graph)
**Output:** $List[V]$: The list of new nodes, representing predicates, in the new graph
**Output:** $V_o$: The output node which depends on the some other nodes (in case that recursive is
        not enabled in $f$) in $List[V]$
$NewNode \leftarrow []$
Determine the lowest possible number of nodes to add $l$ and the highest number of possible
 nodes to add $r$ based on $f$.
$N \leftarrow random(l, r)$;
**while** $i$ *in 1.....N* **do**
    $p \leftarrow A \; randomly \; assigned \; predicate$;
    $G.AddNode(p)$;
    $NewNode.Push(p)$;
**end**
$OutNode \leftarrow RandomSelect(NewNode)$;
**while** $p$ *in NewNode* **do**
    **if** *"recursive" not in $f$ and $p == OutNode$* **then**
        continue;
    **end**
    $G.AddEdge(p, OutNode)$
**end**
**return** *G, NewNode, OutNode*

---

After the structure of the graph is generated, we are going to attach rule information to each edge of the graph using the Rule Generation algorithm 3. Since we are doing Graph Generation and Rule Generation alternately, in the rule generation we only care about edges that don't already has a rule, we will skip the edges that already has a rule associated with that.

---

**Algorithm 3:** Rule Generation

---

**Input:** $G$: The existing graph
**Parameters:** $f$: The set of features Enabled
**Output:** $G$: The generated graph (including the old information in the existing graph)
$SelectedOp \leftarrow Set()$
$SelectedOp.add(RandomSelect(Boxminus, Boxplus, Diamondplusm, Diamondminus))$
**if** *"mixed_operators" in $f$* **then**
    Randomly select and add more operators to SelectedOp;
**end**
**while** *Edge in G* **do**
    $u, v, a \leftarrow G$;
    **if** *No rule is associated with Edge* **then**
        $Op \leftarrow$ Randomly select an operator from $SelectedOp$;
        $Interval \leftarrow$ Randomly create an interval;
        Create an item literal with $Op$ and $Interval$ and associated that with $Edge$;
    **end**
**end**
**return** $G$

---

## G  COMPUTATIONAL RESOURCE REQUIREMENT

For LLama-3-8B and Qwen2.5-32B, we used two NVIDIA H100 80GB HBM3 GPUs, and hosted using vLLM. Zero-shot and few-shot inference usually take less than 10 mintues, and chain-of-thought usually takes less than 1 hour.

For Distilled DeepSeek Models, we used two NVIDIA H100 80GB HBM3 GPUs, and inference usually takes less than 1 hour.

For DeepSeek R1, we used the cloud inference platform Fireworks AI[4], and the full evaluation takes less than $10 USD.

For GPT-4o, we used the cloud inference platform OpenAI [5]. The full evaluation takes less than $100 USD.

## H  LIMITATION

Our experiments were constrained by the speed, computational resources, and financial costs associated with utilizing GPT-4o and DeepSeek-R1. For instance, although our generator allows for the creation of temporal data and rules with arbitrary sizes, we obtained results across multiple temporal reasoning datasets of varying complexities on a relatively small scale due to the financial costs associated with GPT-4o and DeepSeek-R1 API calls.

Another limitation of this preliminary exploration into testing the temporal reasoning abilities of LLMs is that we present experimental results from only three prompting settings, despite the availability of more advanced prompting strategies. Additionally, while our results demonstrate that DeepSeek-R1 and its distilled models significantly outperform the other evaluated models, we do not establish the underlying factors contributing to this superiority. Our human analysis of certain error cases provides limited insights, and we do not propose an effective method for enhancing LLMs' ability to handle temporal logic reasoning problems.

---

[4]Fireworks AI Platform can be accessed at `https://fireworks.ai/`
[5]The OpenAI Platform can be accessed at `https://platform.openai.com/`

