# OpenReview forum: "t-BEN: A Temporal Logic Guided Approach for Temporal Reasoning Benchmark Generation"
_ICLR.cc/2026/Conference — Submitted to ICLR 2026_

### Official Review · Reviewer_VJ8G · 2025-10-30

**Soundness:** 3
**Presentation:** 3
**Contribution:** 2
**Rating:** 4
**Confidence:** 3

**Summary:**

This paper introduces T-BEN, a new benchmark to test how well LLMs handle temporal logic reasoning. It works by automatically generating fresh, logic-based problems that models have not seen during training, which is a good way to test true reasoning instead of memorization. The most interesting finding is that DeepSeek-R1 is exceptionally good at these tasks, while many other top models, including GPT-4o, perform poorly, often no better than random guessing.

**Strengths:**

- It maybe the first benchmark of its kind, filling a major gap in how we evaluate formal reasoning in LLMs.
- The method is solid. Generating new problems on the fly is smart, and the tiered difficulty (from simple to recursive rules) is sound for finding the breaking points of different models.

**Weaknesses:**

- The benchmark focuses only on one specific type of logic (DatalogMTL). The findings might not apply to other important temporal logics used in fields like formal verification.
- The paper suggests why DeepSeek-R1 is so good (its training method), but this is treated as a hypothesis without direct experimental proof.
- The problems are abstract and machine-generated. It is unclear how well performance on T-BEN would translate to messy, real-world tasks that require temporal understanding (e.g., interpreting medical records or financial reports).

**Questions:**

1. The DeepSeek-R1 result is fascinating. Can you elaborate further on why you believe its training strategy is the key factor? Are there any other plausible explanations for its unique ability?
2. How do you see the skills tested in T-BEN transferring to practical, real-world applications? Is there a clear path from solving these abstract puzzles to solving real problems?
3. Have you considered extending this framework to other temporal logics like LTL or CTL? This could help determine if DeepSeek's ability is specific to DatalogMTL or represents a more general strength in formal reasoning.

---

> ### Author Response · Authors · 2025-11-19
>
> > The DeepSeek-R1 result is fascinating. Can you elaborate further on why you believe its training strategy is the key factor? Are there any other plausible explanations for its unique ability?
>
> Thank you for raising this question. Although we cannot make strong causal claims here, our experimental results show that DeepSeek-R1 and other reasoning models consistently outperform non-reasoning models. This suggests that for tasks like ours—where temporal logic queries are fully synthetic and never observed during pretraining—models benefit from explicitly learning to produce intermediate reasoning steps.
>
> In particular, for DeepSeek-R, our hypothesis is that its reinforcement-learning optimization over reasoning trajectories encourages the model to maintain coherent intermediate states, follow multi-step deductive paths, and systematically explore solution spaces. These capabilities align closely with the demands of temporal logic reasoning and likely contribute to its strong performance.
>
> > How do you see the skills tested in tBen transferring to practical, real-world applications? Is there a clear path from solving these abstract puzzles to solving real problems?
>
> Thank you for the question. Although tBen uses abstract synthetic tasks, the temporal reasoning skills it evaluates—such as reasoning over event order, duration, and rule-based temporal constraints—are foundational to many real-world applications (e.g., planning, scheduling, monitoring, and multi-step agent decision-making). Our aim is to assess whether models possess these core abilities in a clean, contamination-free setting. We see strong performance on tBen as a necessary step toward, though not sufficient for, robust temporal reasoning in practical applications.
>
> > Have you considered extending this framework to other temporal logics like LTL or CTL? This could help determine if DeepSeek's ability is specific to DatalogMTL or represents a more general strength in formal reasoning.
>
> Thank you for the suggestion. DatalogMTL, LTL, and CTL all belong to the family of temporal logics. For example, consider the expression “active power was below 0.15 MW (P) for a period of one minute.” In LTL, this can be expressed as $\bigcirc_{p}P  \wedge \bigcirc_{p}^{2}P   \wedge \cdots  \wedge \bigcirc_{p}^{60}P  $ whereas in DatalogMTL, it can be more compactly represented as $\Box_{[1,60]}$. Both representations capture the same temporal semantics.  In this paper, our goal is to isolate LLMs’ capacity for reasoning over temporal knowledge, rather than their ability to interpret different forms with the same temporal logic semantics. Hence, our current work focuses on DatalogMTL because it provides a clean, rule-based, and data-centric formalism that aligns well with the types of fact-based temporal reasoning we aim to evaluate. This allows us to generate controllable complexity while maintaining precise semantics.

---

> > ### Author Response · Authors · 2025-11-24
> > **Kind Request for Review of Our Responses**
> >
> > Dear Reviewer VJ8G,
> >
> > Thank you so much again for your great efforts in reviewing our paper!  As the deadline for the discussion is fast approaching, **we sincerely hope that our replies have adequately addressed your feedback and will encourage you to reconsider the rating.**
> >
> > If you have any further questions, please do not hesitate to discuss them with us before the discussion deadline, and  we are really looking forward to having a discussion with you on the OpenReview system. Your insights are invaluable in refining our paper!
> >
> >
> > Best regards,
> >
> > The authors of Paper 13708

---

### Official Review · Reviewer_7xog · 2025-10-30

**Soundness:** 3
**Presentation:** 3
**Contribution:** 4
**Rating:** 8
**Confidence:** 3

**Summary:**

The paper introduces t-BEN, a synthetic benchmark suite for evaluating LLMs on temporal logic reasoning. Each instance consists of temporal data, rules written in DatalogMTL, and a single fact-entailment query. The benchmark is intentionally semantics-grounded, supports both symbolic and natural-language variants of the same task (via rule verbalization), and is designed to be scalable and data-leakage-resistant through on-the-fly randomized generation. The authors define six rule-complexity “levels” (SingleAtom, MultiAtoms, Rational timeline, MixedOperators, MultiRules, Recursive) and describe a three-stage generator (graph construction, data generation, rule generation) with verification using the MeTeoR reasoner. Experiments across 13 models report strong performance for “reasoning” models (notably o3, Gemini 2.5 variants, DeepSeek-R1) and weaker results for general LLMs (e.g., GPT-4o without CoT); recursive and multi-rule settings are the hardest. The authors include ablations on number of relevant rules, operator variety, and injected distractors, and compare symbolic vs. NL variants (similar difficulty; symbolic slightly easier).

Soundness:
The paper addresses a noticeable lack of symbolic temporal logic reasoning benchmarks for LLMs. While previous benchmarks have attempted to measure formal reasoning abilities of these models in first-order logic, the extent of their capabilities on temporal logic reasoning has yet to be examined. The conceptualization of fact entailment as a fundamental temporal logic reasoning task is a sound choice that enables the synthesis of a broad range of temporal logic reasoning problems. Lastly, the adoption of DatalogMTL over less expressive temporal logic variations encourages future work towards deployable LLM-based temporal logic reasoning systems.

Presentation:
The work is well written and establishes most of the preliminary information needed to contextualize the proposal. The style and notation are largely consistent with related works. It would be nice to include one additional overview figure to better describe dataset generation.

Contribution:
The introduction of a dataset which can measure temporal logic reasoning abilities of language models is a novel and timely contribution to the field of symbolic reasoning integration with language models. As previously stated, there are have been recent efforts in benchmarking various aspects of symbolic reasoning with LLMs, but none have yet covered the domain of (metric) temporal logics.

**Strengths:**

1. Sound choices in problem framing, including the use of DatalogMTL for its expressive semantics and breadth.
2. The benchmark is the first to evaluate temporal logic fact-entailment in a semantics-grounded setting and the first to support symbolic and NL variants tied to the same underlying task.
3. The benchmark generator includes correctness guarantees via MeTeoR. Results cover 13 models, several prompting settings, and detailed ablations (rule depth, distractors, operator counts). Recursive and multi-rule difficulty trends are empirically demonstrated.

**Weaknesses:**

1. The six “levels” are not formally characterized by computational hardness. This is acknowledged, but even a short theoretical appendix benchmarking against known DatalogMTL fragment complexity classes would add rigor.
2. Although symbolic methods can't run on NL prompts, comparing their runtime or success rates for the symbolic subset (vs. LLM success modes) would help position LLMs as complements vs. replacements.

**Questions:**

While the authors acknowledge synthetic focus, the paper does not explore bridging to real-world domains (e.g., planning logs, event-stream QA, formal verification logs). If the author can provide a small illustrative case, this would strengthen applicability.

---

> ### Author Response · Authors · 2025-11-19
>
> > The six “levels” are not formally characterized by computational hardness. This is acknowledged, but even a short theoretical appendix benchmarking against known DatalogMTL fragment complexity classes would add rigor.
>
> We agree that connecting our six levels to the formal complexity-theoretic fragments of DatalogMTL would improve the rigor of the work. However, as we noted in the paper, our levels (SingleAtom, MultiAtoms, Rational, MixedOperators, MultiRules, Recursive) were designed as pragmatic difficulty tiers to capture increasing structural and reasoning complexity for LLMs (e.g., more atoms and operators). They are therefore not intended as formal complexity classes, and in Section 4.1 we explicitly acknowledge that higher levels correspond to increased difficulty because symbolic reasoners like MeTeoR require more “temporal reasoning steps,” rather than because they belong to specific complexity fragments.
>
> That said, we appreciate the reviewer’s suggestion and agree that providing a theoretical correspondence would enhance the rigor of our work. In particular, prior work—Tractable Fragments of Datalog with Metric Temporal Operators (IJCAI 2020)—has established data-complexity results for several DatalogMTL fragments. While our levels are not exactly fragments defined in the literature, we think they roughly correspond as below:
>
> | tBen     | Closest Known DatalogMTL Fragment (IJCAI 2020)              | Known Data Complexity          |
> |----------------|--------------------------------------------------------------|--------------------------------|
> | SingleAtom     | Core, non-conjunctive                                        |  TC⁰-complete                 |
> | MultiAtoms     | Linear fragment                     | NL-complete                    |
> | Rational       | Core fragment with general metric operators                  | NL-complete                      |
> | MixedOperators | Practical Core fragment (restricted operators)               |    PTIME              |
> | MultiRules     | Non-recursive conjunctive fragment  (≤1 IDB atom per rule)                        | Between PTIME   and PSPACE-complete  |
> | Recursive      | Full DatalogMTL (with recursion)                             | PSPACE-complete                |
>
>
> > Although symbolic methods can't run on NL prompts, comparing their runtime or success rates for the symbolic subset (vs. LLM success modes) would help position LLMs as complements vs. replacements.
>
> Regarding runtime comparison, a direct comparison of reasoning speed or system throughput is not straightforward because LRMs and symbolic reasoners operate under fundamentally different computational paradigms. For instance, symbolic systems typically perform explicit search on CPUs, whereas LRMs rely on parallel GPU inference, making raw speed comparisons difficult to interpret. Regarding success rates, the symbolic reasoner used in this paper (MeTeoR) guarantees strict correctness for each sample, i.e., it achieves 100% accuracy in this setting.
>
> We note that the goal of this paper is to evaluate whether LLMs can correctly execute temporal reasoning tasks. Unlike symbolic reasoners, which are typically language-specific (e.g., MeTeoR for DatalogMTL or NuSMV for LTL), LLMs are more generalizable.  Their ability to perform correct temporal reasoning is therefore a necessary prerequisite for assessing whether LLMs can serve as complements to—or substitutes for—symbolic reasoners.
>
> > While the authors acknowledge synthetic focus, the paper does not explore bridging to real-world domains (e.g., planning logs, event-stream QA, formal verification logs). If the author can provide a small illustrative case, this would strengthen applicability.
>
> Thank you for the suggestion. While tBen is designed as a synthetic, contamination-free benchmark to isolate temporal reasoning capabilities, we agree that connecting these skills to real-world domains is important. As a small illustrative case, we consider an example from Siemens’ remote diagnostic system described in the paper https://arxiv.org/pdf/1703.08982 (page 5)
> . A service engineer may be interested in active power trips of a turbine (ActivePowerTrip), defined as events where the active power exceeds 1.5 MW for at least 10 seconds, followed within 3 seconds by a period of at least 1 minute where the active power is below 0.15 MW. This scenario can be expressed in DatalogMTL as:
>
>
> $\mathrm{ActivePowerTrip}(v) \leftarrow \mathrm{Turbine}(v) \wedge \Box_{[0,1m]} \,\mathrm{ActivePowerBelow0.15}(v) \wedge \Diamond_{[60s\,,63s]}\Box_{[0,10s]},\mathrm{ActivePowerAbove1.5}(v)$
>
> Given temporal observations of the turbine, correctly determining whether an active power trip occurred requires the model to perform accurate temporal reasoning over the provided event data in conjunction with the DatalogMTL rules. This demonstrates that the temporal reasoning skills assessed by tBen could be directly applicable to practical, real-world tasks.

---

> > ### Author Response · Authors · 2025-11-24
> > **Kind Request for Review of Our Responses**
> >
> > Dear Reviewer 7xog,
> >
> > Thanks so much again for your great efforts in reviewing our paper!  Your initial comments are very insightful and helpful in refining our paper. As the deadline for the discussion is fast approaching, we sincerely hope that our replies have adequately addressed your feedback. If you have any further questions, please do not hesitate to discuss them with us before the discussion deadline, and  we are really looking forward to having a discussion with you on the OpenReview system.
> >
> >
> > Best regards,
> >
> > The authors of Paper 13708

---

### Official Review · Reviewer_kxkL · 2025-11-01

**Soundness:** 3
**Presentation:** 2
**Contribution:** 2
**Rating:** 4
**Confidence:** 4

**Summary:**

This paper proposes a synthetic symbolic dataset for temporal reasoning based on the language DatalogMTL, an extension of Datalog with metric temporal logic. Rules in the dataset are classified into six types. The paper evaluates the performance of seven LLMs with this dataset, and results show that LLMs including GPT-4o exhibit poor performance, but LRMs including DeepSeek-R1 deliver strong results.

**Strengths:**

1. The proposed dataset can potentially serve as a testbed for improving LLMs' temporal reasoning abilities.

2. Experimental results show that LRMs may serve as alternatives to traditional symbolic reasoners.

**Weaknesses:**

1. Although the dataset also has a natural language form, it is generated with a template-based approach, and hence lacks the linguistic complexity of real-world natural language. Thus the paper sheds limited insight on LLMs' abilities of  temporal reasoning with natural language.

2. Although experimental results show that LRMs may serve as alternatives to traditional symbolic reasoners, the paper does not do an experimental comparative analysis between the temporal reasoning abilities of LRMs and symbolic reasoners, for example, in terms of reasoning speed. Thus it is not clear in what sense LRMs can serve as complementary tools to symbolic reasoners.

3. Despite the weak performance of LLMs on this dataset, the paper does not discuss possible ways to improve the temporal reasoning abilities of LLMs.

4. Related work section should focus on closely related works to make clear the contributions of this paper, namely works on evaluating LLMs' temporal reasoning abilities such as (Wang & Zhao 2024), (Xiong et al. 2024) and (Fatemi et al. 2025). However, the discussion of these closely related works is very limited.

With the above limitations, I feel the significance of the proposed dataset is unclear.

**Questions:**

Could you comment on the issues raised in the limitations part?

---

> ### Author Response · Authors · 2025-11-19
>
> Thank you for your comments!
>
> > Although the dataset also has a natural language form, it is generated with a template-based approach, and hence lacks the linguistic complexity of real-world natural language. Thus the paper sheds limited insight on LLMs' abilities of temporal reasoning with natural language.
>
> We agree that using templates to convert symbolic examples into natural-language expressions may yield less linguistic diversity than real-world text. However, this is an intentional trade-off given the complexity of tBen’s temporal and relational structures. Allowing more varied natural-language realizations (e.g., via LLM-based rewriting or paraphrasing) would introduce a substantial risk of semantic drift and inaccuracies. The template-based approach ensures that all inputs remain semantically precise, eliminating noise from conversion errors. Our goal in this work is to isolate LLMs’ capacity for reasoning over temporal knowledge, rather than their ability to interpret natural language.
>
> > Although experimental results show that LRMs may serve as alternatives to traditional symbolic reasoners, the paper does not do an experimental comparative analysis between the temporal reasoning abilities of LRMs and symbolic reasoners, for example, in terms of reasoning speed. Thus it is not clear in what sense LRMs can serve as complementary tools to symbolic reasoners.
>
> Thank you for raising this point.  Our experimental results demonstrate only that LRMs can serve as alternatives or complements to symbolic reasoners with respect to their ability to perform temporal reasoning on the benchmark tasks—not to a full system-level comparison involving runtime or engineering considerations. However, a direct comparison of reasoning speed or system throughput is not the focus of this work because:
> - LRMs and symbolic reasoners operate under fundamentally different computational paradigms, making raw speed comparisons difficult to interpret (e.g., symbolic systems run on CPU with explicit search, whereas LRMs utilize parallel GPU inference).
>
> - The primary goal  of this paper is to evaluate whether LLMs can correctly understand and execute temporal primitives and their compositions, which we believe is a very important prerequisite for assessing their potential substitutability for, or complementarity with, symbolic reasoners.
>
> > Despite the weak performance of LLMs on this dataset, the paper does not discuss possible ways to improve the temporal reasoning abilities of LLMs.
>
> We would like to clarify that the primary focus of this paper is to introduce a contamination-free testbed, grounded in temporal logic, for evaluating the temporal reasoning capabilities of current models. While we do not propose concrete methods for improving LLMs’ temporal reasoning abilities, our extensive experiments across 13 models provide valuable insights into the strengths and limitations of both LLMs and LRMs in temporal reasoning tasks, which we believe can help inform future research on model improvements.
>
> > Related work section should focus on closely related works to make clear the contributions of this paper, namely works on evaluating LLMs' temporal reasoning abilities such as (Wang & Zhao 2024), (Xiong et al. 2024) and (Fatemi et al. 2025). However, the discussion of these closely related works is very limited.
>
> Thank you for bringing these two papers to our attention. Although we have already cited both works—Wang & Zhao (2024) in Line 047 and Fatemi et al. (2025) in Line 136—we agree that providing a more detailed discussion of their contributions will benefit readers, and we will incorporate this in the revised version. However, we note that each of these works examines aspects of LLM temporal reasoning that differ from the goals of tBen:
>
> - TRAM (Wang & Zhao, 2024) consists entirely of multiple-choice questions and does not require LLMs to reason over long chains of temporal facts. Its construction method also differs significantly from ours: TRAM reuses existing static datasets (e.g., MC-TACO, SQuAD) and programmatically generates distractors and permutations using templates.
>
> - ToT (Fatemi et al., 2025) adopts a graph-based synthetic data-generation approach, varying graph size, question type, and fact order to control task complexity.
>
> In contrast, our work is strictly grounded in temporal logic, a formalism extensively studied in the logic-based AI literature. The complexity of our questions arises from variations in rule structures, temporal operators, and event durations. To the best of our knowledge, tBe is the first benchmark to systematically evaluate LLMs’ formal temporal reasoning under a precise logical framework, filling an important gap in existing evaluation methodologies.

---

> > ### Author Response · Authors · 2025-11-24
> > **Kind Request for Review of Our Responses**
> >
> > Dear Reviewer kxkL,
> >
> > Thank you so much again for your great efforts in reviewing our paper!  We have addressed all your questions in detail. As the deadline for the discussion is fast approaching, **we sincerely hope that our replies have adequately addressed your feedback and will encourage you to reconsider the rating.**
> >
> > If you have any further questions, please do not hesitate to discuss them with us before the discussion deadline, and we are really looking forward to having a discussion with you on the OpenReview system. Your insights are invaluable in refining our paper!
> >
> > Best regards,
> >
> > The authors of Paper 13708

---

### Author Response · Authors · 2025-12-03
**Rebuttal Summary by Authors**

Dear Area Chair and Reviewers,

We thank all reviewers for their insightful and constructive feedback. We would like to re-emphasize that, unlike existing temporal-reasoning benchmarks, tBen serves as a **dynamic** testbed for evaluating LLM performance on temporal logic reasoning. This dynamic nature offers unique advantages: adjustable difficulty levels that accommodate both stronger and weaker LLMs with  **no data leakage**. Below, we list some strengths highlighted by multiple reviewers:

- All three reviewers **kxkL**, **7xog**, **VJ8G** see t-BEN as a useful benchmark/testbed that fills a major gap in evaluating formal/temporal reasoning in LLMs, providing a novel and timely way to measure these abilities.

- Reviewer **7xog**, **VJ8G** mention that the benchmark is the first semantics-grounded temporal logic reasoning benchmark, addressing the noticeable lack of such datasets for LLMs.

- Reviewer **7xog**, **VJ8G** agree that the benchmark's tiered difficulty is a sensible way to capture increasing structural/reasoning complexity and to identify breaking points of different models.

We now address two common concerns raised by  reviewers:

> t-BEN may not reflect realistic language or realistic temporal-reasoning tasks, so the external validity is unclear.

This design choice is intentional: our primary goal in this work is to cleanly isolate temporal reasoning ability, not to model the full complexity of open-ended language understanding. Using controlled templates ensures that each natural-language instance remains semantically faithful to its underlying DatalogMTL rules, avoiding the semantic drift and annotation noise that can arise from free-form or paraphrase-based generation. This tight alignment between symbolic and NL variants is crucial for attributing model failures to reasoning rather than to misinterpretation of ambiguous language. At the same time, the temporal competencies we test are directly relevant to many practical domains, as illustrated by our mapping of an industrial turbine "active power trip" scenario into DatalogMTL. In this sense, t-BEN should be viewed as a contamination-free, semantics-grounded substrate for core temporal reasoning skills that can later be composed with richer language understanding in downstream applications.

> There are limited comparisons with symbolic reasoners

In our setting, the symbolic backbone (MeTeoR) is used precisely because it guarantees strict correctness on all instances, effectively achieving 100% accuracy on t-BEN; this makes a success-rate comparison trivial and shifts the central question to whether learning-based models can approach this standard. In addition, LRMs and symbolic systems operate under fundamentally different computational paradigms (explicit search on CPUs vs. highly parallel neural inference on GPUs), so wall-clock comparisons are difficult to interpret without substantial engineering assumptions. Our focus here is more foundational: to determine whether LLMs and LRMs can at all execute the kinds of temporally structured reasoning steps that symbolic reasoners handle by design.

---

### Meta-Review · Area_Chair_8WaE · 2026-01-04

**Summary:**

The paper introduces t-BEN, a synthetic benchmark for temporal logic reasoning based on DatalogMTL. While the goal of creating a contamination-free, logically grounded testbed is sound in principle, the execution suffers from significant limitations that undermine its practical utility and longevity as a benchmark for the ICLR community.

1. Lack of linguistic diversity and realism: As noted by Reviewers kxkL and VJ8G, the reliance on template-based generation results in natural language queries that are rigid and artificial. Real-world temporal reasoning—such as interpreting medical records, legal contracts, or financial reports—involves significant linguistic ambiguity, noise, and varied phrasing. By strictly adhering to templates to avoid "semantic drift," the authors have created a benchmark that tests a model's ability to parse specific synthetic structures rather than its ability to reason about time in natural language. Consequently, the external validity of the benchmark is low; high performance on t-BEN is unlikely to translate to robust temporal reasoning in realistic user applications.

2. Limited scope and challenge: The benchmark focuses exclusively on DatalogMTL. As pointed out by Reviewer VJ8G, this narrows the scope significantly compared to other logical formalisms (like LTL or CTL) used in formal verification. Furthermore, your observation regarding the difficulty of these problems is supported by the empirical results: "Large Reasoning Models" like DeepSeek-R1 already demonstrate strong performance on these tasks. This suggests that the underlying symbolic rules are sufficiently restricted that they do not pose a significant challenge to the emerging class of reasoning-enhanced models. A benchmark that is effectively "solved" or highly tractable for current frontier reasoning models upon release offers limited value for driving future progress.

**Reviewer Concerns:**

Concerns solved: Comparison with Symbolic Reasoners (Reviewers kxkL, 7xog): Reviewers asked for speed/performance comparisons against symbolic solvers. The authors clarified that symbolic solvers (like MeTeoR) are used to generate the ground truth (achieving 100% accuracy by definition), making accuracy comparisons trivial. They also argued that runtime comparisons are invalid due to different hardware paradigms (CPU search vs. GPU inference).

Concerns not solved: Linguistic Diversity & Realism (Reviewers kxkL, VJ8G): The benchmark relies on templates, resulting in rigid, artificial natural language. While the authors successfully argued this is necessary to prevent "semantic drift" and isolate reasoning (a justification accepted by the AC), the benchmark inherently lacks the ambiguity and complexity of real-world text.

**Reviewer Scores:**

I don't think reviewer scores were to change.

---

### Decision · Program_Chairs · 2026-01-26

Reject